# Single VDGA-Based Mixed-Mode Universal Filter and Dual-Mode Quadrature Oscillator

**DOI:** 10.3390/s22145303

**Published:** 2022-07-15

**Authors:** Natchanai Roongmuanpha, Worapong Tangsrirat, Tattaya Pukkalanun

**Affiliations:** School of Engineering, King Mongkut’s Institute of Technology Ladkrabang (KMITL), Bangkok 10520, Thailand; natchanai.roo@gmail.com (N.R.); tattaya.pu@kmitl.ac.th (T.P.)

**Keywords:** voltage differencing gain amplifier (VDGA), mixed-mode, dual-mode, universal biquadratic filter, voltage-mode (VM), current-mode (CM), trans-admittance-mode (TAM), trans-impedance-mode (TIM), quadrature oscillator

## Abstract

This article presents the circuit designs for a mixed-mode universal biquadratic filter and a dual-mode quadrature oscillator, both of which use a single voltage differencing gain amplifier (VDGA), one resistor, and two capacitors. The proposed circuit has the following performance characteristics: (i) simultaneous implementation of standard biquadratic filter functions with three inputs and two outputs in all four possible modes, namely, voltage-mode (VM), current-mode (CM), trans-admittance-mode (TAM), and trans-impedance-mode (TIM); (ii) electronic adjustment of the natural angular frequency and independently single-resistance controllable high-quality factor; (iii) performing a dual-mode quadrature oscillator with simultaneous voltage and current output responses; (iv) orthogonal resistive and/or electronic control of the oscillation condition and frequency; (v) employing all grounded passive components in the quadrature oscillator function; and (vi) simpler topology due to the use of a single VDGA. VDGA non-idealities and parasitic elements are also investigated and analyzed in terms of their influence on circuit performance. To prove the study hypotheses, computer simulations with TSMC 0.18 μm CMOS technology and experimental confirmatory testing with off-the-shelf integrated circuits LM13600 have been performed.

## 1. Introduction

Universal filters are useful active filters that permit all the five typical biquadratic filter functions simultaneously, namely low-pass (LP), band-pass (BP), high-pass (HP), band-stop (BS), and all-pass (AP) responses with the same topology. These circuits are frequently used in the design of a wide variety of electronic instruments, data communications, and control systems since they enable the implementation of various filtering functions based on port selections. In many analog signal processing applications, an active mixed-mode universal biquadratic filter (MUBF) with input voltages and/or currents, and output voltages and/or currents, is necessarily required. Over the last decade, numerous universal biquad filter realizations in mixed-mode operations based on different active components have already been developed in [1,2,3,4,5,6,7,8,9,10,11,12,13,14,15,16,17,18,19,20,21,22,23,24,25,26,27,28,29,30,31].

Two periodic waveforms having a 90° phase difference, known as a quadrature oscillator (QO), are frequently required in the design of electronic communication systems. QOs have applications in communication systems to operate quadrature mixers, in instrumentation and measurement systems to test and diagnose electronic devices and circuits, as well as in single-sideband generators. Interesting QO circuits have been reported in the literature, which includes realizations using various active building blocks [32,33,34,35,36,37,38,39,40].

Note that the above-mentioned realizations only work with the mixed-mode universal biquad filter or the QO circuit. Interestingly, [41,42,43,44,45,46,47,48,49,50] suggest circuit realizations that can perform both universal biquad filter and QO with the same circuit design. A comparison of available related works is made in Table 1. A thorough examination of Table 1 shows that the MUBF realizations in [2,6,8,11,16,19,20,24,41,42,43,44,45,46,47,48,49,50] were unable to realize the various filter functions in all four possible operation modes, including voltage-mode (VM), current-mode (CM), trans-admittance-mode (TAM), and trans-impedance-mode (TIM). Furthermore, the QOs in [44,47,49] only generated voltage-mode quadrature signals, while those in [42,43] generated current-mode quadrature signals. In certain modern electronic applications, a QO circuit that provides both voltage and current outputs simultaneously, called a dual-mode QO (DMQO), may be required. However, little effort has been made on the design of DMQO [32,33,34,35,36,37,38,39,40]. The realizations in [1,2,4,5,6,7,9,10,11,12,13,14,15,17,18,19,20,22,23,25,26,29,30,31,33,34,37,39,42,43,44,45,46,47,49,50] need two or more active components. Four or more passive components were used in the circuits described in [2,3,4,6,8,9,12,13,14,15,16,19,21,22,23,24,26,27,29,31,32,34,35,37,39,45,47,48]. Moreover, the inbuilt tunability feature is not provided for the approaches presented in [2,3,4,6,8,9,12,13,19,22,23,31,32,34,39,47]. The circuits in [2,6,7,8,10,13,18,21,23,28,31,40,42,43,48,50] also do not allow for independent tuning of the important characteristics, such as the natural angular frequency and quality factor for MUBF, or the oscillation condition and frequency for QO.

A voltage differencing gain amplifier (VDGA), a recently introduced active element, was introduced in 2013 [51], and has since been used in a variety of analog signal processing applications [52,53,54,55,56]. As previously stated, we found that no attempts have been made to use a single VDGA to implement both MUBF and DMQO with the same configuration. Considering the growing interest in multiple-mode signal processing, this work therefore proposes a MUBF and DMQO circuit designed using a single VDGA. The circuit makes use of only one resistor and two capacitors as passive components. By significantly modifying the design, the proposed circuit can be categorized as either a MUBF or a DMQO. For the proposed mixed-mode filter, it can perform VM, CM, TAM, and TIM biquadratic filters with an orthogonally controlled the natural angular frequency and the quality factor. Furthermore, the high-*Q* filter may be easily implemented with a single resistor. For the proposed DMQO, the oscillation condition and the oscillation frequency are separately programmable. The performance of the proposed MUBF and DMQO circuit is illustrated by PSPICE simulation results. To further demonstrate the practicability of the circuit, the experimental test results using commercially available ICs are also conducted.

## 2. Overview of VDGA

The VDGA device, which was recently described in [51], is a versatile active element. A wide range of VDGA-based analog signal processing solutions, including active universal filters [52,53], quadrature oscillators [54,55], and tunable capacitance multiplier [56], have been developed in the technical literature. Its schematic representation is illustrated in Figure 1, with *p* and *n* representing high-impedance voltage input terminals, *z*+, *z*−, *x*, and *o* representing high-impedance current output terminals, and *w* representing a low-impedance voltage output terminal. The ideal terminal property of the VDGA element is represented by the matrix expression [51]:(1)$$\left[\begin{array}{c} i_{z+}\\ i_{z-}\\ i_{x}\\ \begin{array}{l} v_{w}\\ i_{o} \end{array} \end{array}\right]=\left[\begin{array}{cccc} g_{mA} & -g_{mA} & 0 & 0\\ -g_{mA} & g_{mA} & 0 & 0\\ 0 & 0 & g_{mB} & 0\\ 0 & 0 & \beta & 0\\ 0 & 0 & 0 & -g_{mC} \end{array}\right].\left[\begin{array}{c} v_{p}\\ v_{n}\\ v_{z+}\\ v_{w} \end{array}\right],$$
where *g_mk_* (*k* = *A*, *B*, *C*) is the transconductance gain and *β* is the transfer voltage gain of the VDGA.

Figure 2 shows the probable CMOS-built VDGA internal circuit implementation, which comprises three voltage-controlled floating current sources M_1*k*_–M_9*k*_. Each M_1*k*_–M_9*k*_ implements the corresponding independent adjustable transconductance *g_mk_*, as written below [57]:(2)$$g_{\mathrm{mk}}≅\left(\frac{g_{1k}g_{2k}}{g_{1k}+g_{2k}}\right)+\left(\frac{g_{3k}g_{4k}}{g_{3k}+g_{4k}}\right),$$
where
(3)$$g_{\mathrm{ik}}=\sqrt{\mu C_{\mathrm{ox}}\frac{W}{L}I_{\mathrm{Bk}}},\;(i=1,\;2,\;3,\;4),$$
*μ* is the effective channel electronic mobility, *C_ox_* is the gate-oxide capacitance per unit area, and *W* and *L* are the respective channel width and length of M_1*k*_–M_4*k*_. Because each transconductance *g_ik_* is proportional to the square root of the bias current *I_Bk_*, the value of *g_mk_* may be electronically scaled using Equations (2) and (3). A current-controlled voltage amplifier is also accomplished in Figure 2 by a pair of transconductors M_1B_–M_4B_ and M_1C_–M_4C_ with a voltage gain of *β* = *v_w_*/*v_z+_* = *g_mB_*/*g_mC_*.

## 3. Proposed Mixed-Mode Universal Biquadratic Filter

Figure 3 depicts the proposed universal filter configuration, which consists of a single VDGA, one resistor, and two capacitors. This configuration can be used to implement the mixed-mode universal biquad filter, which includes VM, CM, TAM, and TIM, by selecting appropriate input and output signals, as detailed below.

➣*VM universal biquadratic filter*: With *i_in_* = 0, all the five general voltage-mode biquadratic filter functions for this three-input two-output universal filter can be achieved as follows.With *v_in_* = *v_i_*_3_ (input voltage) and *v_i_*_1_ = *v_i_*_2_ = 0 (grounded), the following LP and BP filter responses are obtained from *v_o_*_1_ and *v_o_*_2_, respectively:
(4)$$T_{VLP}\left(s\right)=\frac{v_{o1}}{v_{in}}=\left(-\frac{1}{g_{mC}R}\right)T_{LP}\left(s\right),$$and
(5)$$T_{VBP}\left(s\right)=\frac{v_{o2}}{v_{in}}=T_{BP}\left(s\right).$$With *v_in_* = *v_i_*_2_ and *v_i_*_1_ = *v_i_*_3_ = 0, the HP response is obtained from *v_o_*_2_, as given by:
(6)$$T_{VHP}\left(s\right)=\frac{v_{o2}}{v_{in}}=T_{HP}\left(s\right).$$
With *v_in_* = *v_i_*_1_ = *v_i_*_2_, and *v_i_*_3_ = 0, the BS response is obtained from *v_o_*_2_, as given by:
(7)$$T_{VBS}\left(s\right)=\frac{v_{o2}}{v_{in}}=T_{BS}\left(s\right).$$
With *v_in_* = *v_i_*_1_ = *v_i_*_2_ = −*v_i_*_3_, the AP response is also obtained from *v_o_*_2_, as given by:
(8)$$T_{VAP}\left(s\right)=\frac{v_{o2}}{v_{in}}=T_{AP}\left(s\right).$$


In Equations (4)–(8), the transfer functions *T_LP_*(*s*), *T_BP_*(*s*), *T_HP_*(*s*), *T_BS_*(*s*), and *T_AP_*(*s*), are as follows.
(9)$$T_{LP}\left(s\right)=\frac{\left(\frac{g_{mA}g_{mB}}{C_{1}C_{2}}\right)}{D\left(s\right)},$$
(10)$$T_{BP}\left(s\right)=\frac{\left(\frac{s}{RC_{2}}\right)}{D\left(s\right)},$$
(11)$$T_{HP}\left(s\right)=\frac{s^{2}}{D\left(s\right)},$$
(12)$$T_{BS}\left(s\right)=\frac{s^{2}+\left(\frac{g_{mA}g_{mB}}{C_{1}C_{2}}\right)}{D\left(s\right)},$$
(13)$$T_{AP}\left(s\right)=\frac{s^{2}-\left(\frac{s}{RC_{2}}\right)+\left(\frac{g_{mA}g_{mB}}{C_{1}C_{2}}\right)}{D\left(s\right)},$$
and
(14)$$D\left(s\right)=s^{2}+\left(\frac{s}{RC_{2}}\right)+\left(\frac{g_{mA}g_{mB}}{C_{1}C_{2}}\right).$$

Equation (4) reveals that the circuit implements the inverted LP filter function with a passband gain of 1/*g_mC_R*, whereas the others, represented by Equations (5)–(8), have a passband gain of unity. For the VM operation, no element-matching requirements are needed.

➣*CM universal biquadratic filter*: The proposed circuit in Figure 3 can be changed into a CM universal biquad with *v_i_*_1_ = *v_i_*_2_ = *v_i_*_3_ = 0. The five generic current-mode biquad transfer functions realized by this configuration are expressed as follows.
(15)$$T_{ILP}\left(s\right)=\frac{i_{o1}}{i_{in}}=T_{LP}\left(s\right),$$
(16)$$T_{IBP}\left(s\right)=\frac{i_{o2}}{i_{in}}=\left(g_{mA}R\right)T_{BP}\left(s\right),$$
and
(17)$$T_{IHP}\left(s\right)=\frac{i_{o3}}{i_{in}}=T_{HP}\left(s\right),$$
where the passband gain of the BP response is equal to *g_mA_R*. Furthermore, the BS response may be realized by simply adding the currents *i_o_*_1_ and *i_o_*_3_ to realize the following current transfer function:(18)$$T_{IBS}\left(s\right)=\frac{i_{o1}+i_{o3}}{i_{in}}=T_{BS}\left(s\right).$$

Similarly, by retaining *g_mA_R* = 1, the AP current response may be obtained by connecting the three currents *i_o_*_1_, *i_o_*_2_, and *i_o_*_3_ to obtain the following transfer function:(19)$$T_{IAP}\left(s\right)=\frac{i_{o1}-i_{o2}+i_{o3}}{i_{in}}=T_{AP}\left(s\right).$$

➣*TAM universal biquadratic filter*: With *v_in_* = *v_i_*_3_, *v_i_*_1_ = *v_i_*_2_ = 0, and *i_in_* = 0, the TAM filter functions are:
(20)$$T_{YLP}\left(s\right)=\frac{i_{o1}}{v_{in}}=\left(\frac{1}{R}\right)T_{LP}\left(s\right),$$
(21)$$T_{YBP}\left(s\right)=\frac{i_{o2}}{v_{in}}=g_{mA}T_{BP}\left(s\right),$$
(22)$$T_{YHP}\left(s\right)=\frac{i_{o3}}{v_{in}}=\left(\frac{1}{R}\right)T_{HP}\left(s\right),$$
(23)$$T_{YBS}\left(s\right)=\frac{\left(i_{o1}+i_{o3}\right)}{v_{in}}=\left(\frac{1}{R}\right)T_{BS}\left(s\right),$$
and
(24)$$T_{YAP}\left(s\right)=\frac{\left(i_{o1}-i_{o2}+i_{o3}\right)}{v_{in}}=\left(\frac{1}{R}\right)T_{AP}\left(s\right).$$

Equation (21) represents the TAM filter function of the BP response with an electronically controlled passband gain of *g_mA_*. The passband gains for the LP, HP, BS, and AP filter responses are equal to 1/*R*. It should be noticed from Equation (24) that, in the case of AP filter realization, a simple element condition, *g_mA_R* = 1, is necessary.

➣*TIM universal biquadratic filter*: According to Figure 3, if *v_i_*_1_ = *v_i_*_2_ = *v_i_*_3_ = 0, the configuration is now operating in TIM universal filter. In this case, the two following TIM responses at voltage outputs *v_o_*_1_ and *v_o_*_2_ can simultaneously be obtained as:
(25)$$T_{ZLP}\left(s\right)=\frac{v_{o1}}{i_{in}}=\left(-\frac{1}{g_{mC}}\right)T_{LP}\left(s\right),$$
and
(26)$$T_{ZBP}\left(s\right)=\frac{v_{o2}}{i_{in}}=RT_{BP}\left(s\right).$$

Equations (25) and (26) express the TIM filter functions of the LP and BP filters with passband gains of (−1/*g_mC_*) and *R*, respectively.

As a consequence, the proposed circuit shown in Figure 3 can be considered as a universal mixed-mode biquadratic filter. The natural angular frequency and the quality factor of this filter are given by [53].
(27)$$\omega_{o}=2\pi f_{o}=\sqrt{\frac{g_{mA}g_{mB}}{C_{1}C_{2}}},$$
and
(28)$$Q=R\sqrt{\frac{g_{mA}g_{mB}C_{2}}{C_{1}}}.$$

It is important to note from Equation (27) that the *ω_o_* can be electronically tuned by changing the transconductances *g_mA_* and *g_mB_*. In addition to Equation (28), the high-*Q* universal filter can be easily realized by tuning the resistor *R* without affecting the characteristic frequency *ω_o_*.

## 4. Proposed Dual-Mode Quadrature Oscillator

In Figure 3, by taking *v_i_*_1_ = *v_i_*_2_ = *v_i_*_3_ = *i_in_* = 0, and connecting terminal *z*− to *x* of the VDGA, the proposed mixed-mode universal biquadratic filter can be worked as a quadrature sinusoidal oscillator. Figure 4 shows the proposed dual-mode quadrature oscillator based on the proposed mixed-mode universal filter in Figure 3. It is worth noting that in this design, all of the passive components are grounded. The characteristic equation of the proposed dual-mode quadrature oscillator in Figure 4 is found as [54]:
(29)$$s^{2}+\left(\frac{1-g_{mA}R}{RC_{2}}\right)s+\left(\frac{g_{mA}g_{mB}}{C_{1}C_{2}}\right)=0.$$

From Equation (29), the oscillation condition (OC) and the oscillation frequency (OF) are evaluated by [55]:(30)$$\mathrm{OC}:\;g_{mA}R=1,$$
and
(31)$$\mathrm{OF}:\;\omega_{osc}=2\pi f_{osc}=\sqrt{\frac{g_{mA}g_{mB}}{C_{1}C_{2}}}.$$

As can be observed from Equations (30) and (31), the OC can be controlled simply by changing the value of a grounded resistor *R* without altering the OF, which can be tuned separately using the transconductance *g_mB_*. As a result, the parameters OC and OF of the proposed quadrature oscillator in Figure 4 are orthogonal controllable.

For sinusoidal steady state, the relationship between the output voltages *v_osc_*_1_ and *v_osc_*_2_ is
(32)$$v_{osc1}=\left(\frac{g_{mA}g_{mB}}{\omega_{osc}g_{mC}C_{1}}\right)e^{j90{}^{\circ}}v_{osc2}.$$

Thus, the proposed circuit produces the two marked voltages *v_osc_*_1_ and *v_osc_*_2_ in quadrature signal.

Also from Figure 4, the output current relations from *i_osc_*_1_ to *i_osc_*_2_ and *i_osc_*_3_ at the OF are found as:(33)$$i_{osc1}=\left(\frac{g_{mB}}{\omega_{osc}C_{1}}\right)e^{j90{}^{\circ}}i_{osc2}=\left(\frac{g_{mA}g_{mB}}{\omega_{{}_{osc}}^{2}C_{1}C_{2}}\right)e^{j180{}^{\circ}}i_{osc3}.$$

According to Equation (33), the phase differences between *i_osc_*_1_ and *i_osc_*_2_, as well as *i_osc_*_1_ and *i_osc_*_3_, are 90° and 180°, respectively. This demonstrates that the three output currents are not only 90° out of phase, but also 180° out of phase.

## 5. Non-Ideal Analyses

This section investigates the impact of VDGA non-idealities on the performance of the proposed mixed-mode universal biquad filter and dual-mode quadrature oscillator. In fact, the non-idealities of the VDGA arise mostly from two significant consequences. The first set of effects is caused by finite tracking errors, whereas the second group is caused by the existence of all VDGA terminal parasitics.

### 5.1. Effect of Finite Tracking Errors

Considering the tracking errors of the VDGA into account, the terminal property given by Equation (1) may be reformulated as [53,54,55]:(34)$$\left[\begin{array}{c} i_{z+}\\ i_{z-}\\ i_{x}\\ \begin{array}{l} v_{w}\\ i_{o} \end{array} \end{array}\right]=\left[\begin{array}{cccc} \alpha_{A}g_{mA} & -\alpha_{A}g_{mA} & 0 & 0\\ -\alpha_{A}g_{mA} & \alpha_{A}g_{mA} & 0 & 0\\ 0 & 0 & \alpha_{B}g_{mB} & 0\\ 0 & 0 & \delta \beta & 0\\ 0 & 0 & 0 & -\alpha_{C}g_{mC} \end{array}\right].\left[\begin{array}{c} v_{p}\\ v_{n}\\ v_{z+}\\ v_{w} \end{array}\right],$$
where *α_k_* (*α_k_* = 1 − *ε_α_*) and *δ* (*δ* = 1 − *ε_δ_*) denote the transconductance inaccuracy parameter and the parasitic voltage gain of the VDGA, respectively. These non-ideal parameters differ from unity due to the transfer errors *ε_α_* (|*ε_α_
*| << 1) and *ε_δ_* (|*ε_δ_
*| << 1).

In presence of the VDGA tracking defects, the expressions for the parameters *ω_o_* and *Q* of the proposed mixed-mode universal biquad filter in Figure 3 are modified as:(35)$$\omega_{o}=\sqrt{\frac{\alpha_{A}\alpha_{B}g_{mA}g_{mB}}{C_{1}C_{2}}},$$
and
(36)$$Q=R\sqrt{\frac{\alpha_{A}\alpha_{B}g_{mA}g_{mB}C_{2}}{C_{1}}}.$$

Through the tracking error effects, the values of *ω_o_* and *Q* clearly depart slightly from their ideal values. These variations may be accommodated by altering the transconductance gains *g_mA_* and *g_mB_* via the bias currents of VDGA.

For the proposed dual-mode quadrature oscillator in Figure 4, the modified OC and OF can be derived as:(37)$$\mathrm{OC}:\;\alpha_{A}g_{mA}R=1,$$
and
(38)$$\mathrm{OF}:\;\omega_{osc}=\sqrt{\frac{\alpha_{A}\alpha_{B}g_{mA}g_{mB}}{C_{1}C_{2}}}.$$

It is evident that the non-ideal factors clearly cause the OC and OF parameters to deviate slightly. However, the OC and OF can still be altered through adjusting *R* and *g_mB_*, respectively.

### 5.2. Effect of Parasitics

The non-ideal behavior model of the VDGA including finite parasitic impedances at each terminal is represented in Figure 5. These parasitics consist of resistance in parallel with capacitance for the *p*, *n*, *z*+, *z*−, *x* and *o* terminals, and serial resistance at the *w* terminal [53,54,55]. Because of the presence of these undesired parasitics, the circuit performance may differ from ideality. As a result, the suggested circuits in Figure 3 and Figure 4, including the VDGA parasitics, must be thoroughly examined.

Using the non-ideal model of VDGA shown in Figure 5, the non-ideal *ω_o_* and *Q* of the filter configuration in Figure 3 are found as:(39)$$\omega_{o}=\sqrt{\frac{g_{mA}g_{mB}+\frac{1}{R^{\prime }R_{z+}}}{C_{1}^{\prime }C_{2}^{\prime }}},$$
and
(40)$$Q=\left(\frac{R^{\prime }R_{z+}C_{1}^{\prime }}{R_{z+}C_{1}^{\prime }+R^{\prime }C_{2}^{\prime }}\right)\sqrt{\frac{\left(g_{mA}g_{mB}+\frac{1}{R^{\prime }R_{z+}}\right)C_{2}^{\prime }}{C_{1}^{\prime }}}.$$
where *R′* = *R* ∥ *R_n_* ∥ *R_x_*,
$C_{1}^{\prime }\;=\;C_{1}\;+\;C_{z+}$, and $C_{2}^{\prime }\;=\;C_{2}\;+\;C_{n}\;+\;C_{x}$.

Similarly, the non-ideal OC and OF of the oscillator configuration in Figure 4 are also found as:(41)$$\mathrm{OC}:\;\left(g_{mA}-\frac{C_{2}^{\prime \prime }}{R_{z+}C_{1}^{\prime }}\right)R^{\prime \prime }=1,$$
and
(42)$$\mathrm{OF}:\;\omega_{osc}=\sqrt{\frac{g_{mA}g_{mB}+\left(\frac{1-g_{mA}R^{\prime \prime }}{R^{\prime \prime }R_{z+}}\right)}{C_{1}^{\prime }C_{2}^{\prime \prime }}}.$$
where *R″* = *R* ∥ *R_n_* ∥ *R_z_*_–_ ∥ *R_x_* and $C_{2}^{\prime \prime }\;=\;C_{2}\;+\;C_{n}\;+\;C_{x-}\;+\;C_{x}$. From Equations (39)–(42), the frequency characteristics of the proposed filter and oscillator circuits would be unaffected, if the following constraints were fulfilled:maximum *R* << parasitic resistances (*R_n_*, *R_z_*_–_, *R_x_*),(43)and
minimum (*C*_1_, *C*_2_) >> parasitic capacitances (*C_n_*, *C_z_*_+_, *C_z_*_–_, *C_x_*).(44)

## 6. Simulation Results

In this section, a PSPICE simulation program was carried out to demonstrate the performance of the proposed configurations in Figure 3 and Figure 4. The VDGA was simulated using the CMOS circuit of Figure 2 with TSMC 0.18 μm transistor parameters, and with symmetrical supply voltages of ±0.9 V. Table 2 illustrates the aspect ratios of the CMOS transistors employed for the VDGA circuit in Figure 2. The capacitor settings for global simulations were *C*_1_ = *C*_2_ = 50 pF.

### 6.1. Simulation Verifications of the Proposed Mixed-Mode Universal Filter

The suggested mixed-mode universal filter in Figure 3 was performed with *g_mA_* = *g_mB_* = *g_mC_* = 1 mA/V (*I_BA_* = *I_BB_* = *I_BC_* = 80 μA), and *R* = 1 kΩ, to actualize all the four-mode universal filter responses with *f_o_* = 3.18 MHz and *Q* = 1. Figure 6, Figure 7, Figure 8 and Figure 9 illustrate the simulated and theoretical frequency responses of VM, CM, TAM, and TIM filters, respectively. The disparity between simulated and theoretical gain responses in the HP filters of VM, CM, and TAM, as well as the BP filter of TIM, is greater in the low-frequency range of roughly 1 kHz to 100 kHz. This phenomenon may be explained by the fact that the input or output signals of the circuits were sensed with *C*_2_ and *R*, introducing an undesirable pole that caused significant deviations in low operating frequencies. In Figure 6c and Figure 7c, the phase shifting between input and output signals was measured as −190.70° and −192.74°, respectively, and the gain response was 0.84 dBV and 1.067 dBA down from zero for the frequency ranges varying from 1 kHz to 1 MHz. The simulated *f_o_* and corresponding percentage errors are given in Table 3. It is to be observed that all simulation results are found to be in good consistent with the theoretical values. Figure 10, Figure 11, Figure 12 and Figure 13 depict the transient responses of the proposed filter to the following input signals: (i) a 3.18 MHz sinusoidal input voltage signal with an amplitude of 100 mV (peak-to-peak) applied to the VM and TAM filters; and (ii) 3.18 MHz sinusoidal input current signal with an amplitude of 100 μA (peak-to-peak) applied to the CM and TIM filters. Table 4 shows the total harmonic distortions (THDs) and DC components of the VM, CM, TAM, and TIM outputs in Figure 10, Figure 11, Figure 12 and Figure 13. As can be seen, the THD value is less than 1.92% in all four modes. Thus, there is no significant distortion in the biquad design. The entire power consumption of the circuit was 1.31 mW at ±0.9 V biased voltages.

In addition, the orthogonal tunability of a high-*Q* value for a BP filter in VM is shown in Figure 14. The filter is designed to operate at *f_o_* = 3.18 MHz with *g_mA_* = *g_mB_* = *g_mC_* = 1 mA/V. By simply adjusting the *R* value for 0.5 kΩ, 10 kΩ, and 50 kΩ, the BP responses with various *Q* values of 0.5, 10, and 50 are achieved, respectively. Based on the measured data, the *Q* value was evaluated as 0.495, 8.273, and 44.25, respectively. The relative variation of the *Q* factor remained less than 12%, even when *Q* reached 50.

The effect of temperature variation on the filter parameters is now being investigated. For this purpose, the proposed filter was simulated under ambient temperature changes ranging from 0 to 100 °C with a step of 25 °C. Figure 15 demonstrates the gain and phase variations of the AP filter in VM operation. The findings reveal that, for different temperatures, the gain and phase at *f_o_* vary from −0.44 to −0.5 dBV and from −172 to −223°, respectively.

### 6.2. Simulation Verifications of the Proposed Dual-Mode Quadrature Oscillator

Based on previous component settings, the simulated quadrature voltages *v_osc_*_1_ and *v_osc_*_2_ of the proposed dual-mode quadrature oscillator in Figure 4 are displayed in Figure 16. Figure 16a shows the steady-state waveforms of *v_osc_*_1_ and *v_osc_*_2_, whereas Figure 16b presents the frequency spectrums of the oscillation output voltages. As per the findings, the simulated *f_osc_* was found to be 2.76 MHz, and the phase shift between *v_osc_*_1_ and *v_osc_*_2_ was 85.76°. The attenuations at the second harmonic for *v_osc_*_1_ and *v_osc_*_2_ were 30.30 dBm and 31.45 dBm, respectively. Further, the percentage of THD was 2.46% for *v_osc_*_1_ and 4.28% for *v_osc_*_2_.

Similarly, the simulated steady-state responses and the corresponding frequency spectrums of *i_osc_*_1_, *i_osc_*_2_, and *i_osc_*_3_ are also given in Figure 17. The phase shifts between *i_osc_*_1_ and *i_osc_*_2_, and *i_osc_*_1_ and *i_osc_*_3_ were measured to be 92.73° and 177.82°, respectively. The second-harmonic attenuations for *i_osc_*_1_, *i_osc_*_2_, and *i_osc_*_3_ were 30.05 dBμ, 30.88 dBμ, and 30.87 dBμ, respectively, while the percentage THDs of *i_osc_*_1_, *i_osc_*_2_, and *i_osc_*_3_ were 3.86%, 4.16%, and 3.50%, respectively.

Due to the VDGA transconductance gain *g_mk_* is tuned by the bias current *I_Bk_*, the *f_osc_* of the proposed circuit is a current tunable function. Figure 18 demonstrates the calculation and simulation results for the variations of *f_osc_* as a function of *I_B_*, where *I_B_* = *I_BA_* = *I_BB_* = *I_BC_*.

## 7. Experimental Results

### 7.1. Experimental Verifications of the Proposed Mixed-Mode Universal Filter

To further support the theory, the suggested circuits in Figure 3 and Figure 4 were experimentally verified. As shown in Figure 19, the VDGA was built-in hardware utilizing off-the-shelf IC dual-OTA LM13600s from National Semiconductor [58]. To bias the LM13600, DC supply voltages of ±5 V were employed. A prototype hardware setup for verification purposes of the proposed circuit is illustrated in Figure 20. The component values were set as follows: *g_mA_* = *g_mB_* = *g_mC_* = 1 mA/V (*I_BA_* = *I_BB_* = *I_BC_* = 50 μA), *R* = 1 kΩ, and *C*_1_ = *C*_2_ = 680 pF, actually results in *f_o_* = 234 kHz, and *Q* = 1. In order to measure the input signals for the CM and TIM, a voltage-to-current converter with IC AD844 [59] and a converting resistor *R_C_* of 1 kΩ was used, as illustrated in Figure 21. In Figure 22, two extra AD844s and *R_C_* were used as a current-to-voltage conversion for output signal measurements in CM and TAM operations.

Figure 23, Figure 24, Figure 25, Figure 26 and Figure 27 show the measurements of the input and output waveforms and the relevant output spectrums for the proposed VM filter with a 20 mV (peak) sinewave input voltage at 234 kHz. The THD values of the LP, BP, HP, BS, and AP output responses were 1.88%, 0.25%, 0.57%, 1.84%, and 2.66%, respectively. As can be seen from Figure 23b, Figure 24b, Figure 25b, Figure 26b and Figure 27b, the spurious-free dynamic range (SFDR) for the cases of LP, BP, HP, BS, and AP were measured at 35.03 dBc, 52.65 dBc, 45.68 dBc, 36.35 dBc, and 34.33 dBc, respectively. Figure 28 also shows the experimental gain-frequency responses of the proposed VM filter. The measured results of *f_o_* of VM, CM, TAM, and TIM were found to be 241.13 kHz (error ~+3%), 227.08 kHz (error ~−2.98%), 227.38 kHz (error ~−2.87%), and 233.32 kHz (error ~−0.33%), respectively. In all cases, the practically observed behavior of the circuit was found to be consistent with the theoretical predictions. The experimental test results, thus, verify the practicability of the suggested design. Nevertheless, one observes that the discrepancy between the theoretical and measured results was originally caused by non-ideal gain and parasitic impedance effects of the LM13600s and AD844s. The stray capacitances generated by the breadboard circuit realization also affect the frequency performance of the circuit in experimental testing.

### 7.2. Experimental Verifications of the Proposed Dual-Mode Quadrature Oscillator

According to the experimental measurements for the proposed dual-mode quadrature oscillator in Figure 4, the oscilloscope output waveforms in time-domain and Lissajous pattern of *v_osc_*_1_ and *v_osc_*_2_ are given in Figure 29. By using the same component values as in the previous filter case, the oscillator was constructed to oscillate at an OF of *f_osc_* = 234 kHz. The *f_osc_* observed was 234.1 kHz, which is extremely close to the theoretical value. The phase angle difference between *v_osc_*_1_ and *v_osc_*_2_ was roughly 95.1°, resulting in an absolute phase deviation of 5.67%. Figure 30 also shows the measured frequency spectrum of the *v_osc_*_1_ output. From the experimental testing, the THD and SFDR values for the output *v_osc_*_1_ were 2.85% and 31.38 dBc, respectively.

For *i_osc_*_1_, *i_osc_*_2_, and *i_osc_*_3_ measurements, the current-to-voltage converter circuit as shown in Figure 22 was also employed. The time-domain waveforms and the corresponding Lissajous figures of the oscillator output currents *i_osc_*_1_ and *i_osc_*_2_, and *i_osc_*_2_ and *i_osc_*_3_ are illustrated in Figure 31 and Figure 32, respectively. The quadrature-phase shifts between *i_osc_*_1_ and *i_osc_*_2_, and *i_osc_*_2_ and *i_osc_*_3_ were 96.1° and 85.3°, respectively, deviating from the calculations by 6.78% and 5.22%. The frequency spectrum of the *i_osc_*_1_ output was also recorded and exhibited in Figure 33, with percentage THD and SFDR values of 2.05% and 34 dBc, respectively. Clearly, the generated waveforms observed in the experimental data validate the quadrature relationship of the suggested quadrature oscillator in both VM and CM.

## 8. Discussion

At this point, we would like to briefly discuss the superiority of the proposed MUBF and DMQO design over similar existing designs in the literature. The following observations are based on Table 1.

In contrast to the topologies in [1,2,3,4,5,6,7,8,9,10,11,12,13,14,15,16,17,18,19,20,21,22,23,24,25,26,27,28,29,30,31,32,33,34,35,36,37,38,39,40], the proposed circuit uses the same topology to perform both MUBF and DMQO, whereas the works referenced only perform MUBF [1,2,3,4,5,6,7,8,9,10,11,12,13,14,15,16,17,18,19,20,21,22,23,24,25,26,27,28,29,30,31] or DMQO [32,33,34,35,36,37,38,39,40]. With regard to the MUBF topologies introduced in [5,7,9,10,12,13,14,15,18,22,23,25,26,27,30], all five biquadratic filter functions are implemented in all four modes of operation. These circuits, however, employ more active components, especially at least two active components, than the suggested circuit. Some have three or more passive components [9,12,13,14,15,22,23,24,25,26,27,28,29,30,31] or DMQO [32,33]. In addition to the filters of [2,6,8,11,16,19,20,24], they are limited to only two modes of operation. Designing with a low component count is a simple technique to reduce the total power consumption of the designed circuit. Even though compact MUBF circuits implemented with a single active element have been proposed in [3,8,16,21,24,27,28,41,48], these biquads still use more passive elements than the MUBF circuit proposed in this work. While the designs described in [4,7,12,14,15,18,19,20,23,26,45,49,50] are interesting, they suffer from the usage of two or more different types of active components, which complicates circuit implementation.

In the QO configurations [47,48], there are floating passive elements that are not encouraged for further integration. Several QO designs operated in either VM [44,47,49] or CM [42,43]. As compared to the proposed DMQO circuit, it not only uses grounded passive elements, but it also provides both voltage and current quadrature outputs simultaneously.

The topologies in [1,2,6,7,8,10,13,18,21,23,28,31,40,42,43,48,50] do not offer independent adjustment of their important parameters, but even the proposed MUBF and DMQO design allows independent parameter modification through transconductance (*g_m_*) or single resistance value. Also in the existing literature [2,3,4,6,8,9,12,13,19,22,23,31,32,34,39,47], an electronic control of various parameters is not available.

As a conclusion, it should be noted that the proposed MUBF and DMQO circuit in this study is capable of fulfilling all of the performance features described above simultaneously and without trade-offs.

## 9. Conclusions

This work proposes a compact mixed-mode universal biquadratic filter and dual-mode quadrature oscillator circuit using a single voltage differencing gain amplifier (VDGA). In this design, a canonical structure with one resistor and two capacitors is employed. The proposed universal biquad filter is able to realize generic second-order filter functions in all four modes of operation, namely, VM, CM, TAM, and TIM. It has the feature of orthogonal control of *ω_o_* and *Q* characteristics, and simultaneously the ability to implement a high-*Q* filter with a single resistance adjustment. The quadrature oscillator, which generates both voltage and current output signals simultaneously, is also feasible by slight modification of the proposed configuration. Both the oscillation condition and the oscillation frequency of the proposed quadrature oscillator are non-interactively controlled. The circuits are subjected to non-ideal analysis, including tracking error and parasitic element effects. The simulation and experimental findings prove that the suggested circuit performs in both the mixed-mode universal biquad filter and the dual-mode quadrature oscillator.

## Figures and Tables

**Figure 1 sensors-22-05303-f001:**
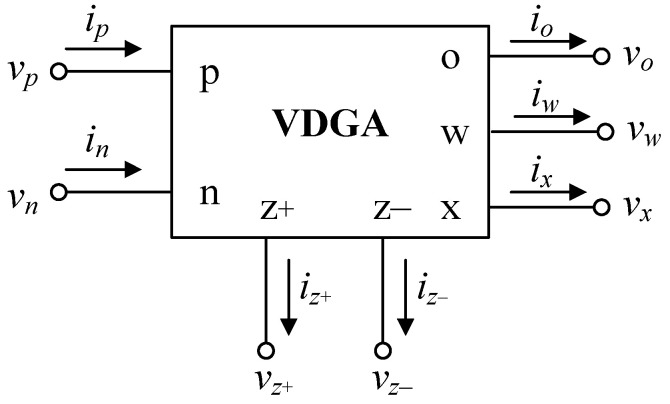
Schematic representation of the VDGA.

**Figure 2 sensors-22-05303-f002:**
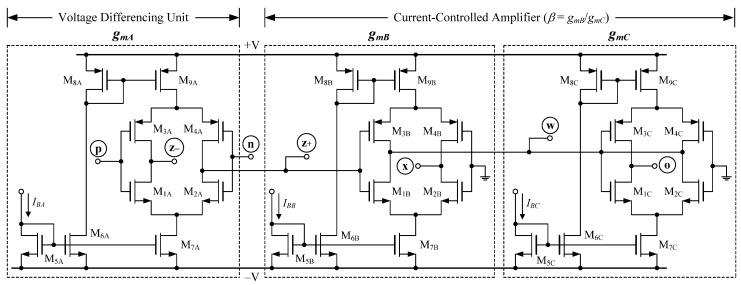
VDGA internal circuit implementation in CMOS technology.

**Figure 3 sensors-22-05303-f003:**
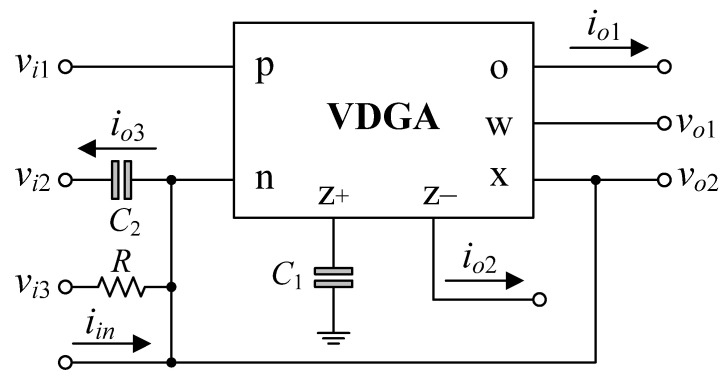
Proposed mixed-mode universal biquadratic filter.

**Figure 4 sensors-22-05303-f004:**
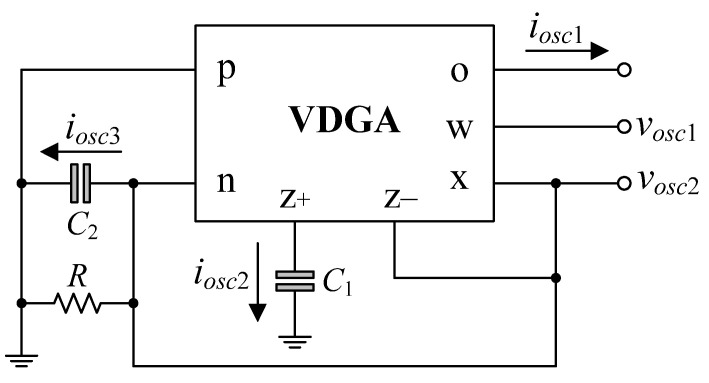
Proposed dual-mode quadrature oscillator.

**Figure 5 sensors-22-05303-f005:**
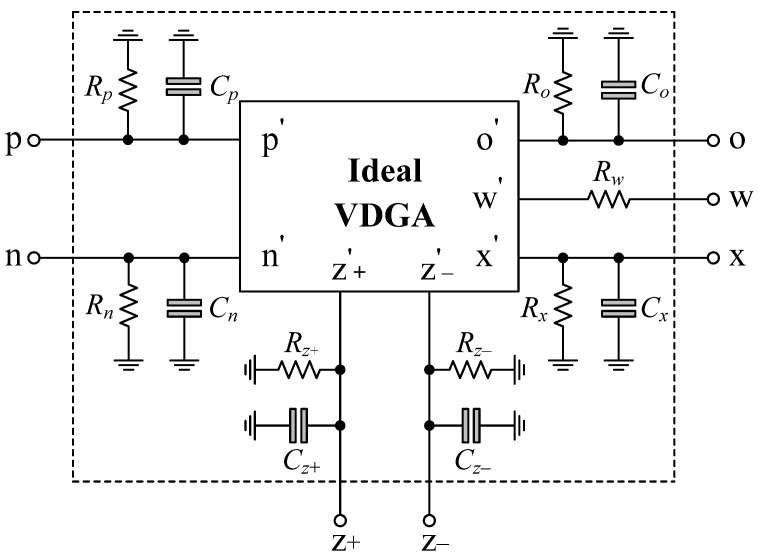
Non-ideal behavior model of the VDGA.

**Figure 6 sensors-22-05303-f006:**
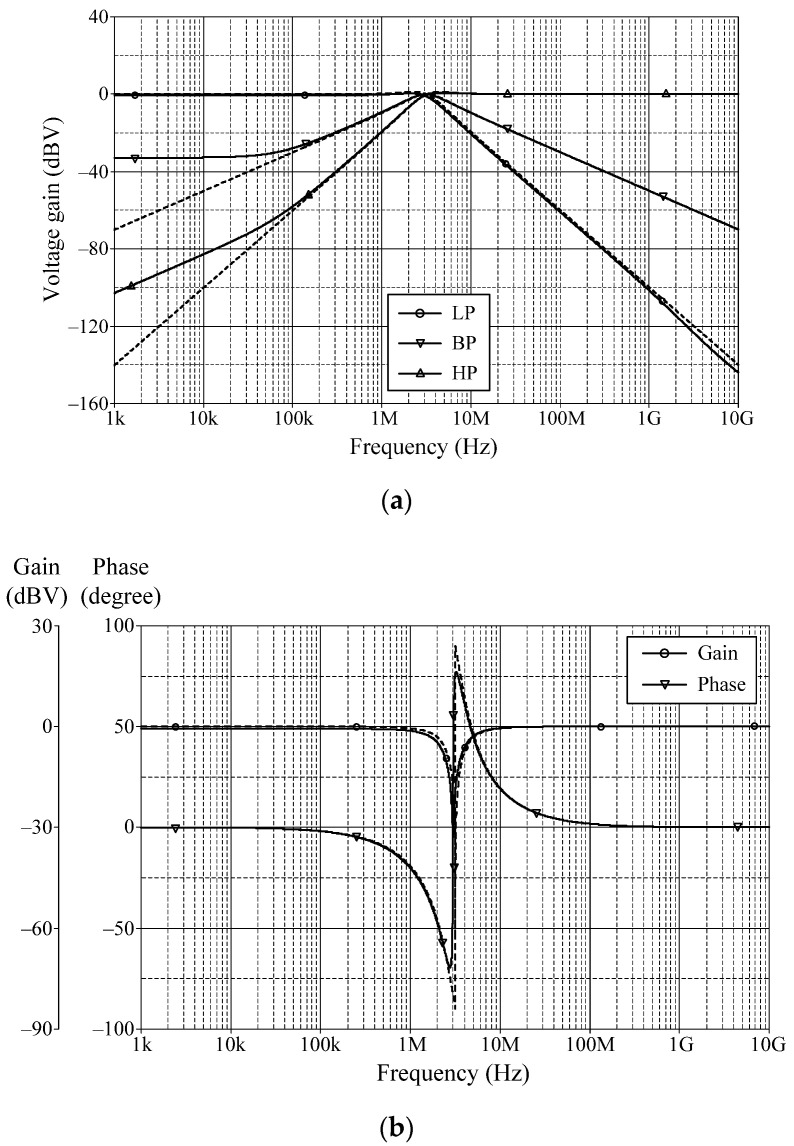
Frequency characteristics for VM (simulated in solid line, theoretical in dashed line): (**a**) LP, BP, and HP; (**b**) BS; (**c**) AP.

**Figure 7 sensors-22-05303-f007:**
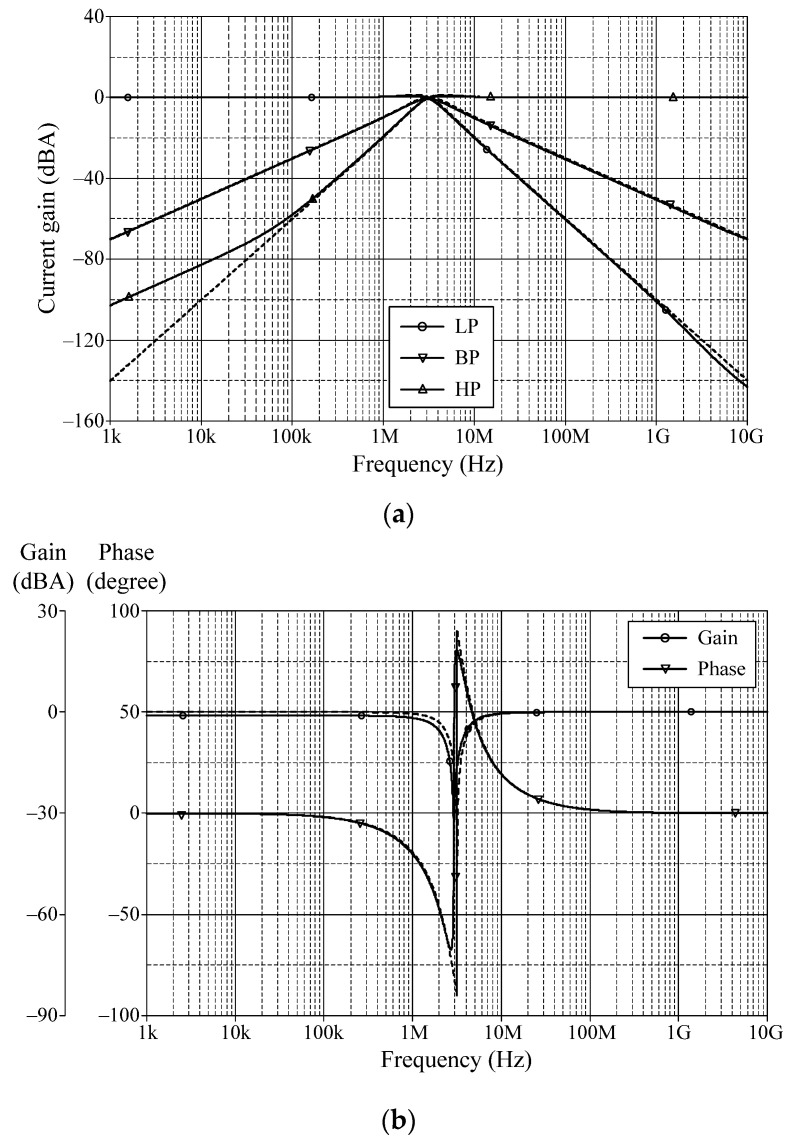
Frequency characteristics for CM (simulated in solid line, theoretical in dashed line): (**a**) LP, BP, and HP; (**b**) BS; (**c**) AP.

**Figure 8 sensors-22-05303-f008:**
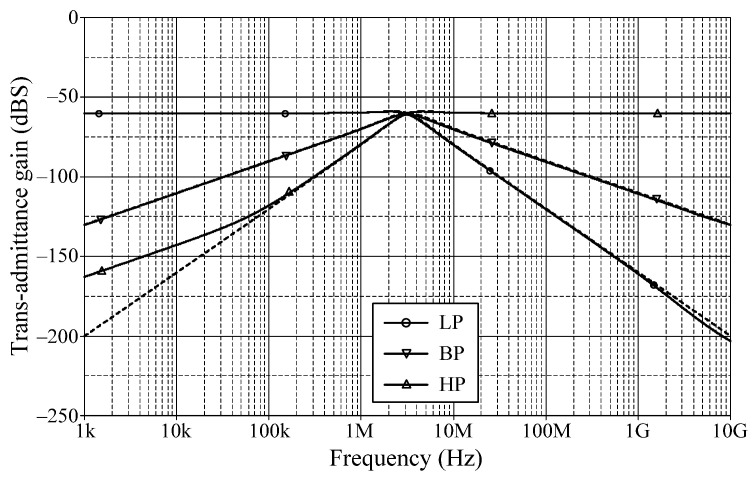
LP, BP, and HP frequency characteristics for TAM (simulated in solid line, theoretical in dashed line).

**Figure 9 sensors-22-05303-f009:**
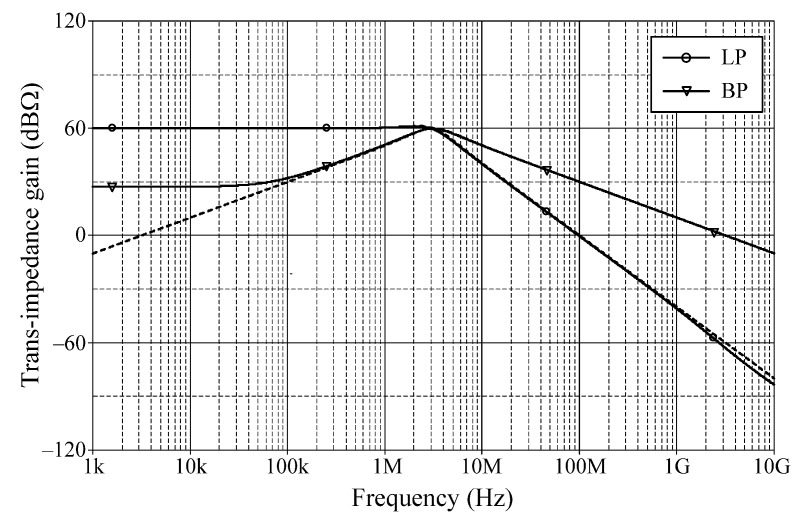
LP and BP frequency characteristics for TIM (simulated in solid line, theoretical in dashed line).

**Figure 10 sensors-22-05303-f010:**
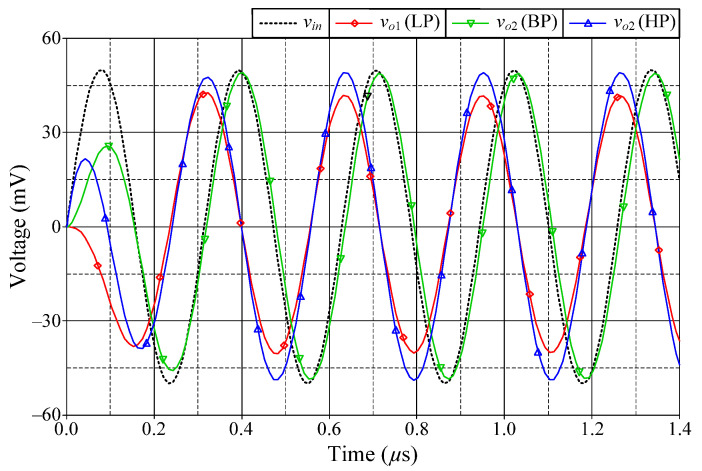
Time-domain responses of the LP, BP, and HP filters in VM.

**Figure 11 sensors-22-05303-f011:**
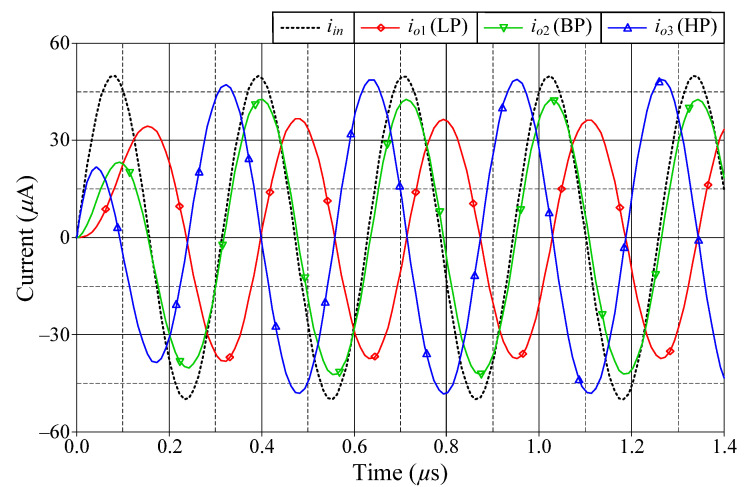
Time-domain responses of the LP, BP, and HP filters in CM.

**Figure 12 sensors-22-05303-f012:**
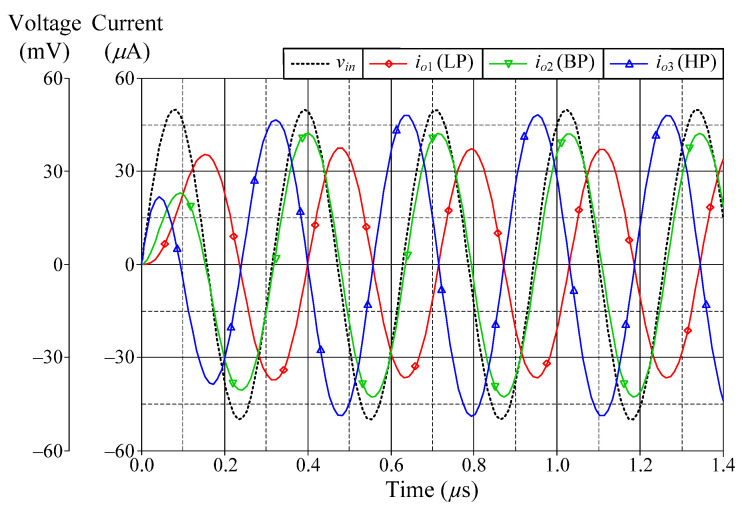
Time-domain responses of the LP, BP, and HP filters in TAM.

**Figure 13 sensors-22-05303-f013:**
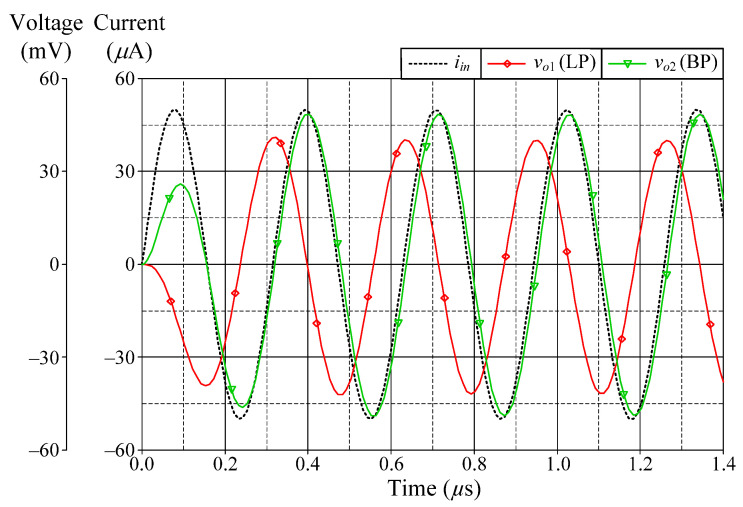
Time-domain responses of the LP and BP filters in TIM.

**Figure 14 sensors-22-05303-f014:**
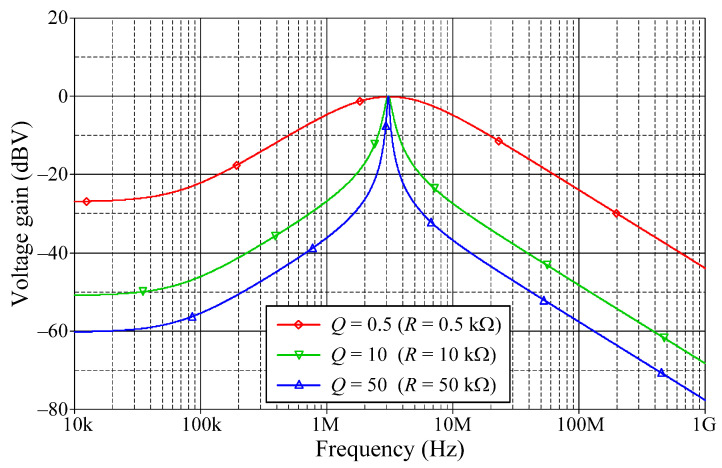
Tunability of *Q* with *f_o_* unchanged for BP filter in VM.

**Figure 15 sensors-22-05303-f015:**
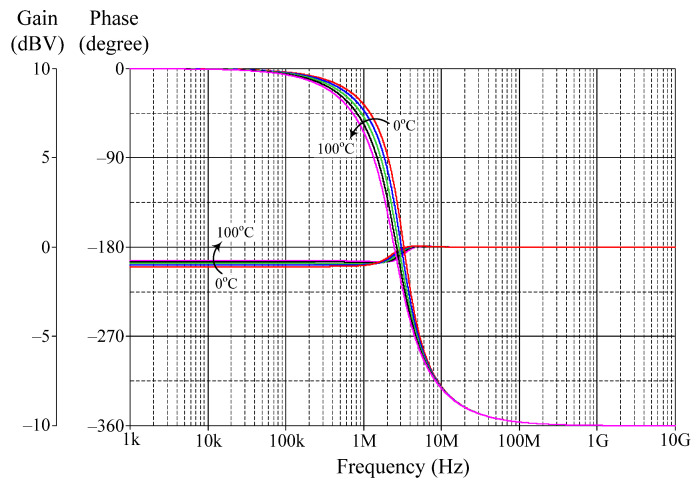
Frequency responses of AP filter in VM for different temperatures (0 °C, 25 °C, 50 °C, 75 °C, and 100 °C).

**Figure 16 sensors-22-05303-f016:**
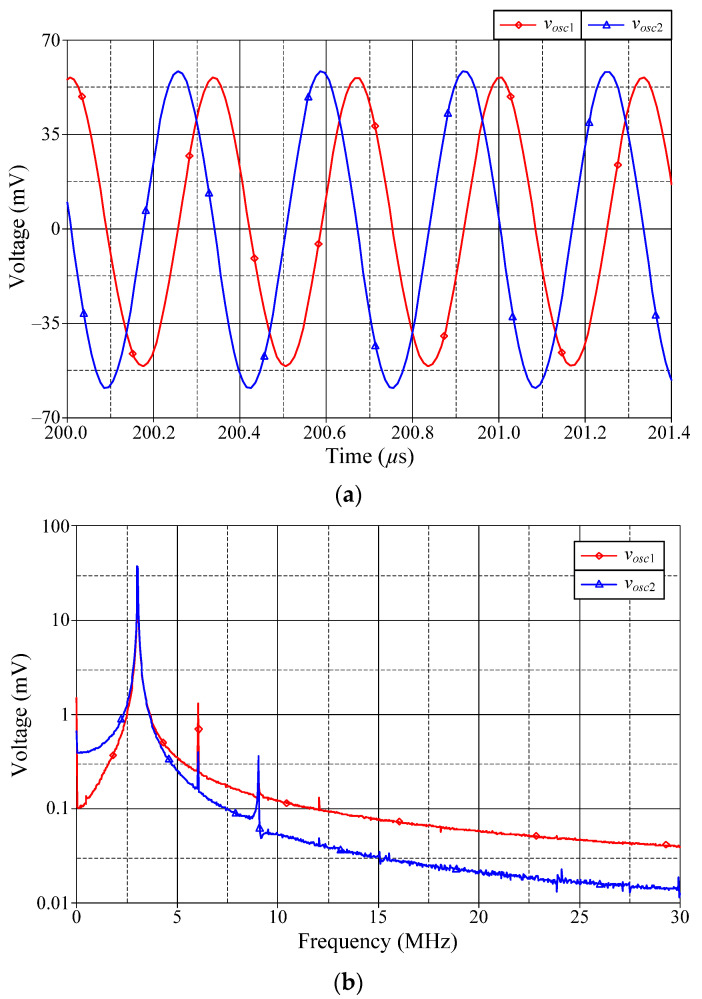
Quadrature output voltages *v_osc_*_1_ and *v_osc_*_2_: (**a**) steady-state waveforms; (**b**) frequency spectrums.

**Figure 17 sensors-22-05303-f017:**
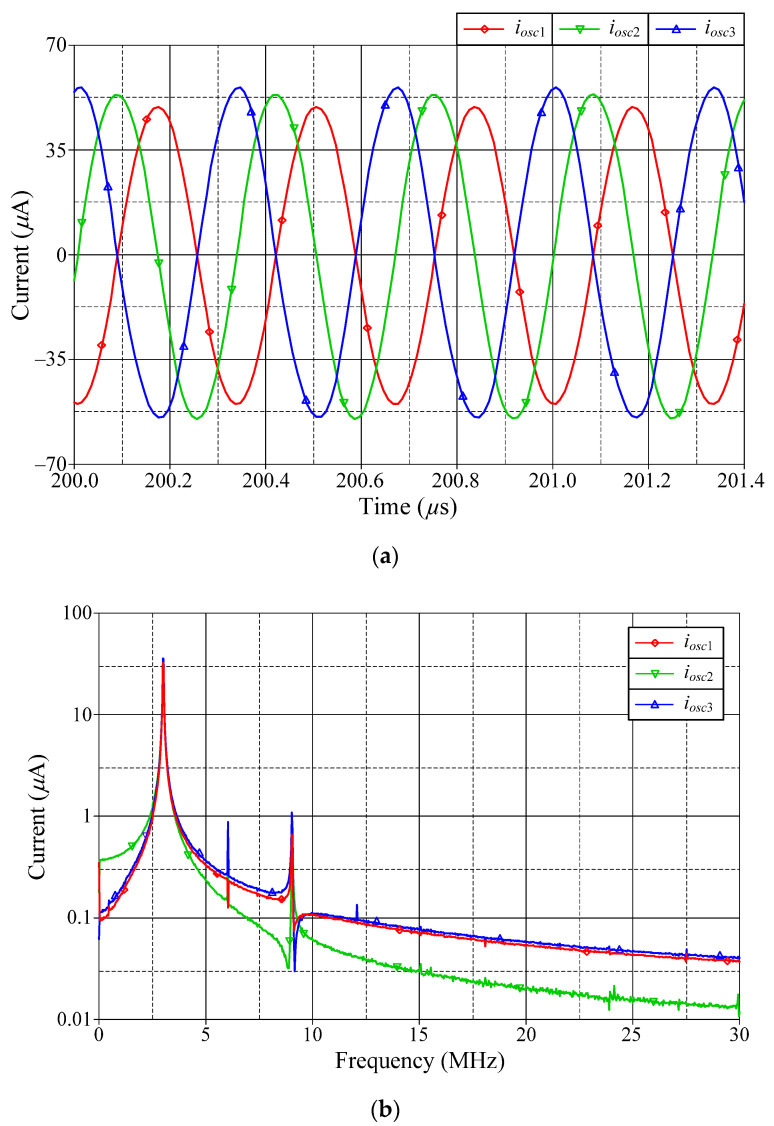
Simulated quadrature output currents *i_osc_*_1_, *i_osc_*_2_, and *i_osc_*_3_: (**a**) steady-state waveforms; (**b**) frequency spectrums.

**Figure 18 sensors-22-05303-f018:**
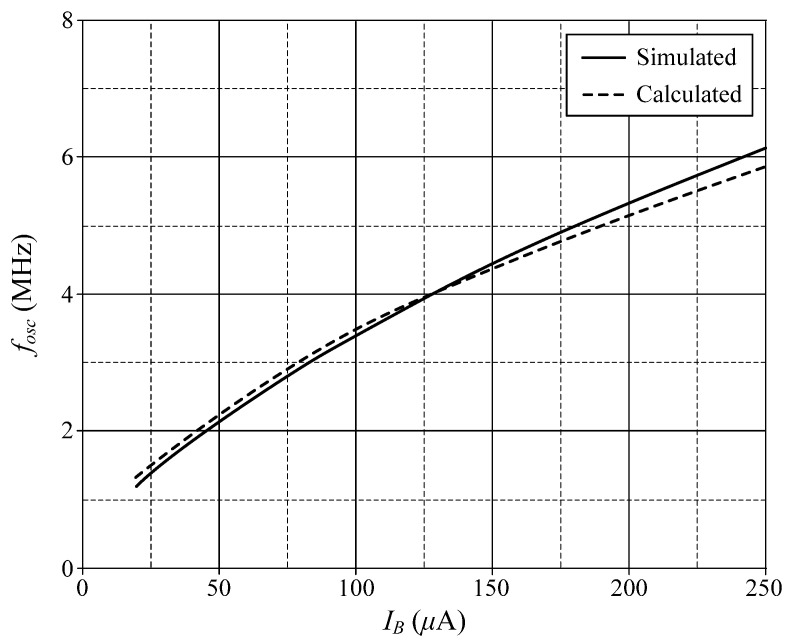
Variations of *f_osc_* against *I_B_* for the proposed quadrature oscillator in Figure 4.

**Figure 19 sensors-22-05303-f019:**
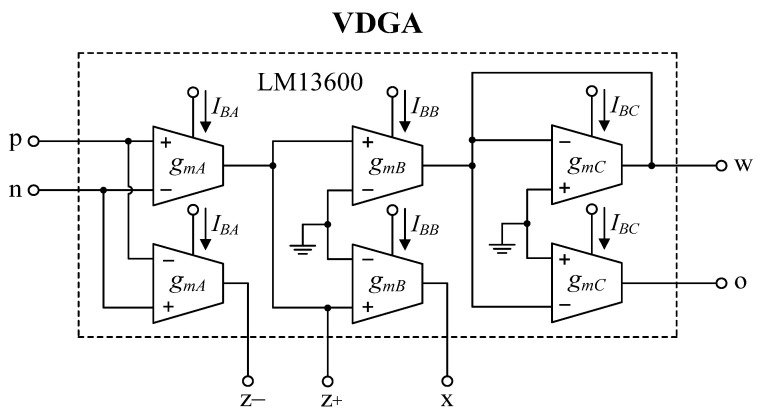
Practical VDGA implementation using off-the-shelf IC LM13600s.

**Figure 20 sensors-22-05303-f020:**
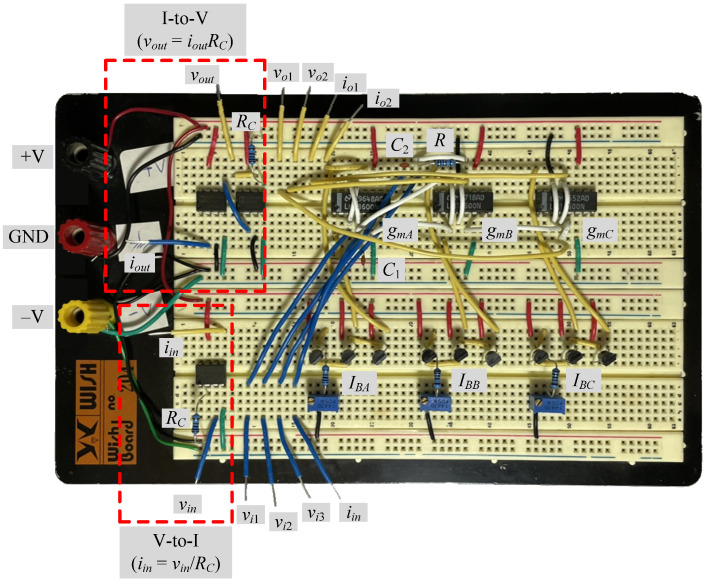
Prototype hardware setup for the experimental verification.

**Figure 21 sensors-22-05303-f021:**
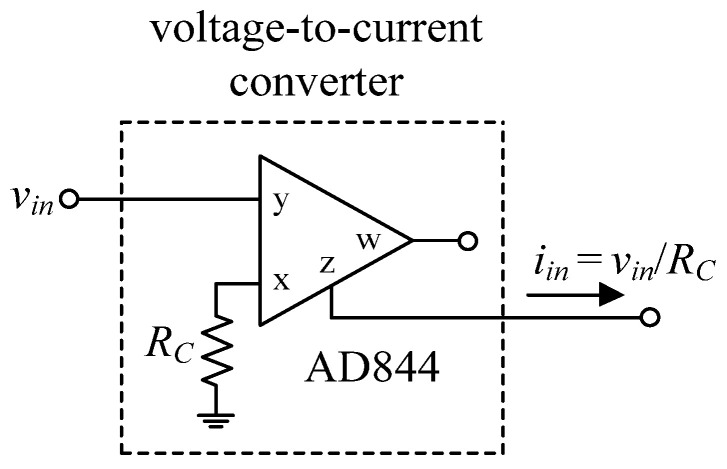
Voltage-to-current conversion for CM and TIM input signal measurements.

**Figure 22 sensors-22-05303-f022:**
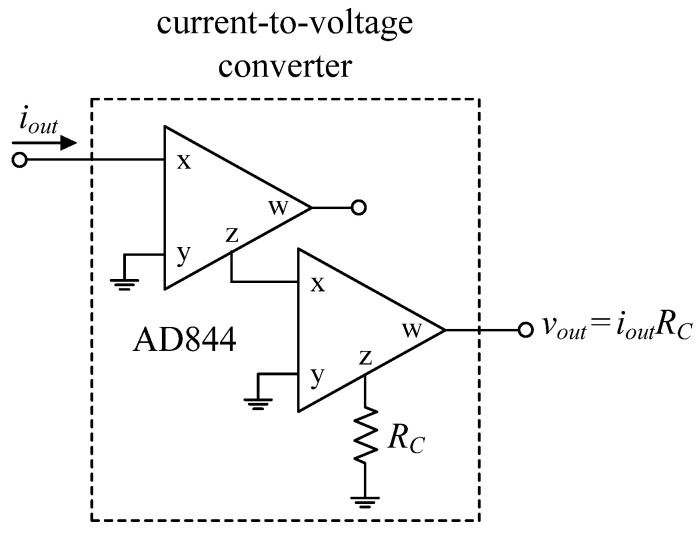
Current-to-voltage conversion for CM and TAM output signal measurements.

**Figure 23 sensors-22-05303-f023:**
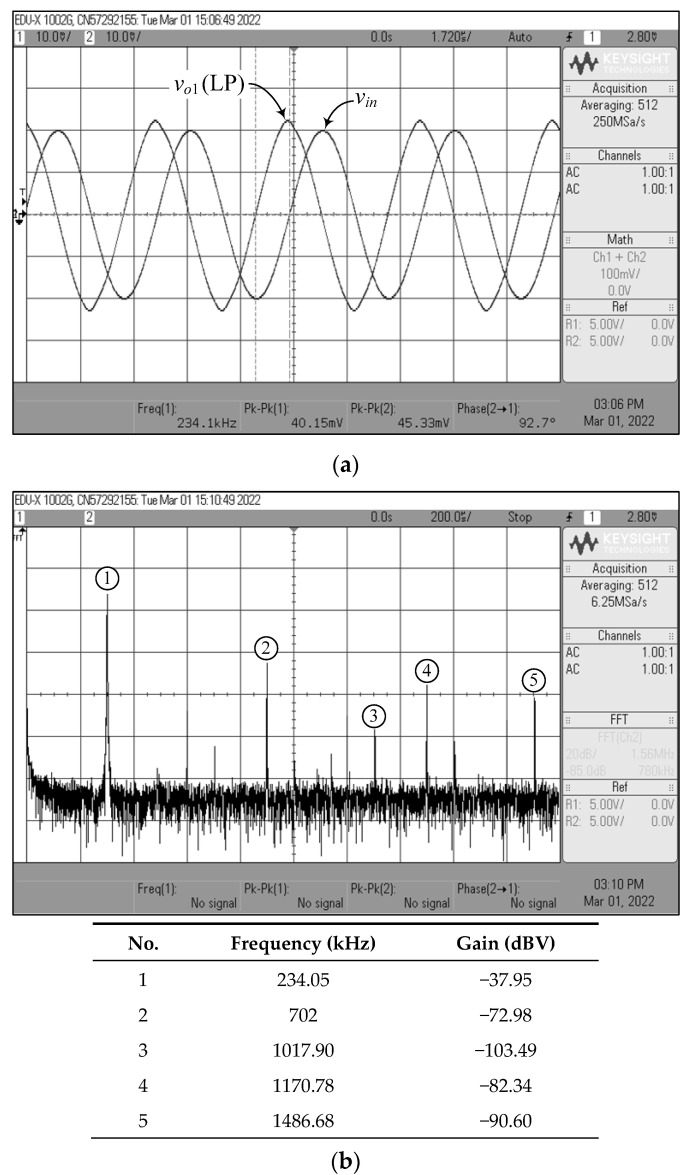
Measured waveforms of the LP filter in VM: (**a**) input and output time responses; (**b**) frequency spectrum.

**Figure 24 sensors-22-05303-f024:**
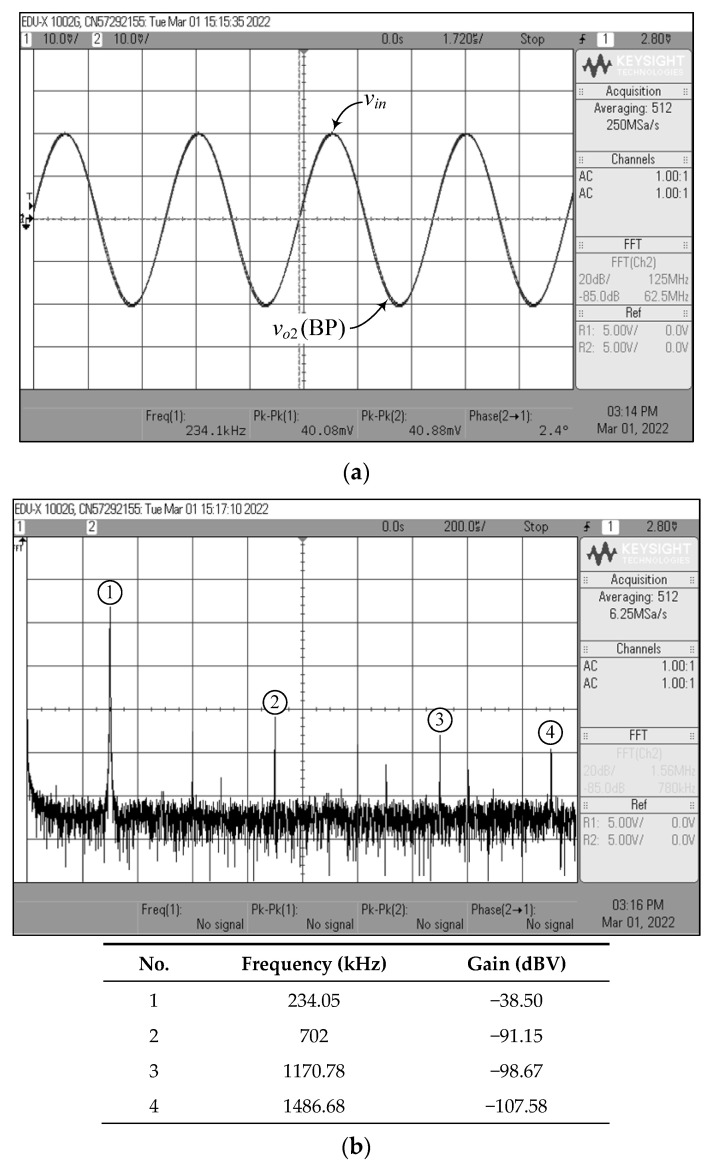
Measured waveforms of the BP filter in VM: (**a**) input and output time responses; (**b**) frequency spectrum.

**Figure 25 sensors-22-05303-f025:**
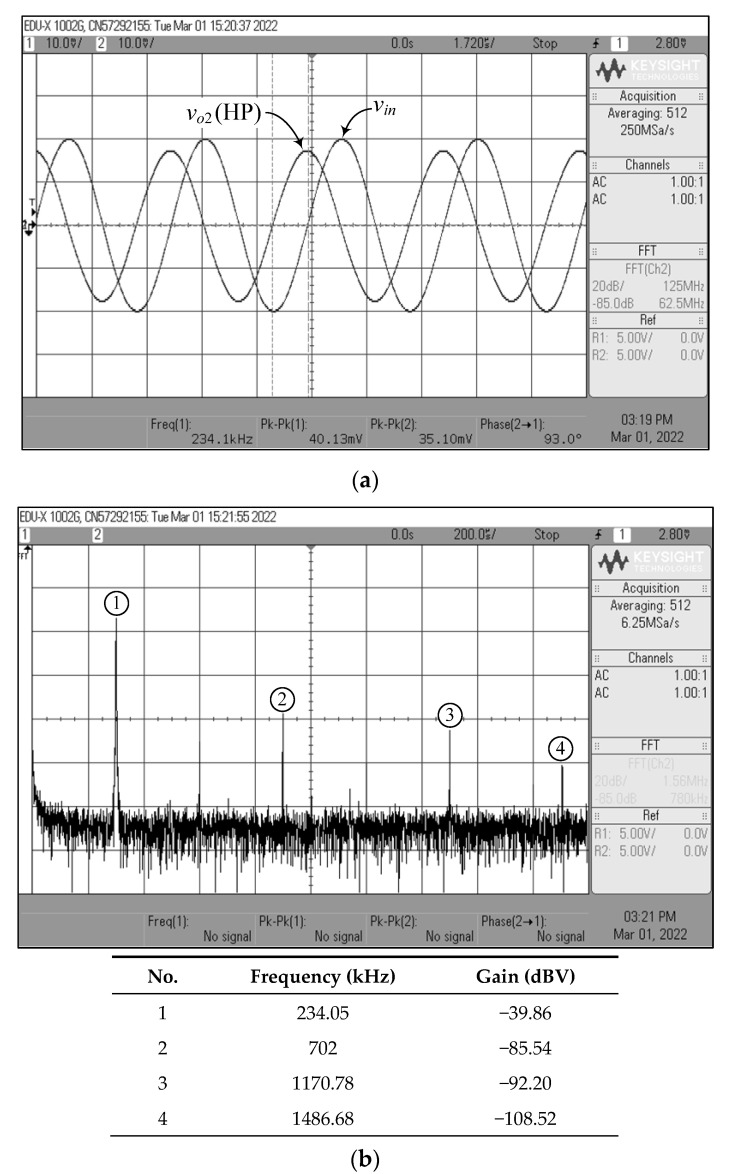
Measured waveforms of the HP filter in VM: (**a**) input and output time responses; (**b**) frequency spectrum.

**Figure 26 sensors-22-05303-f026:**
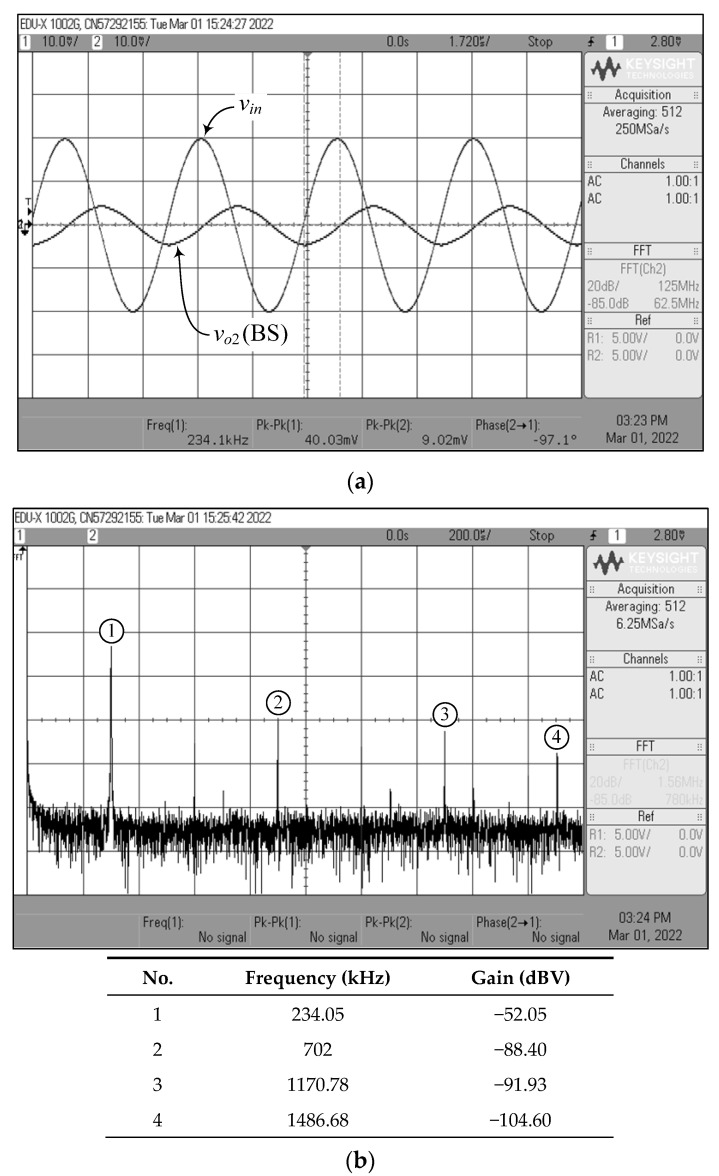
Measured waveforms of the BS filter in VM: (**a**) input and output time responses; (**b**) frequency spectrum.

**Figure 27 sensors-22-05303-f027:**
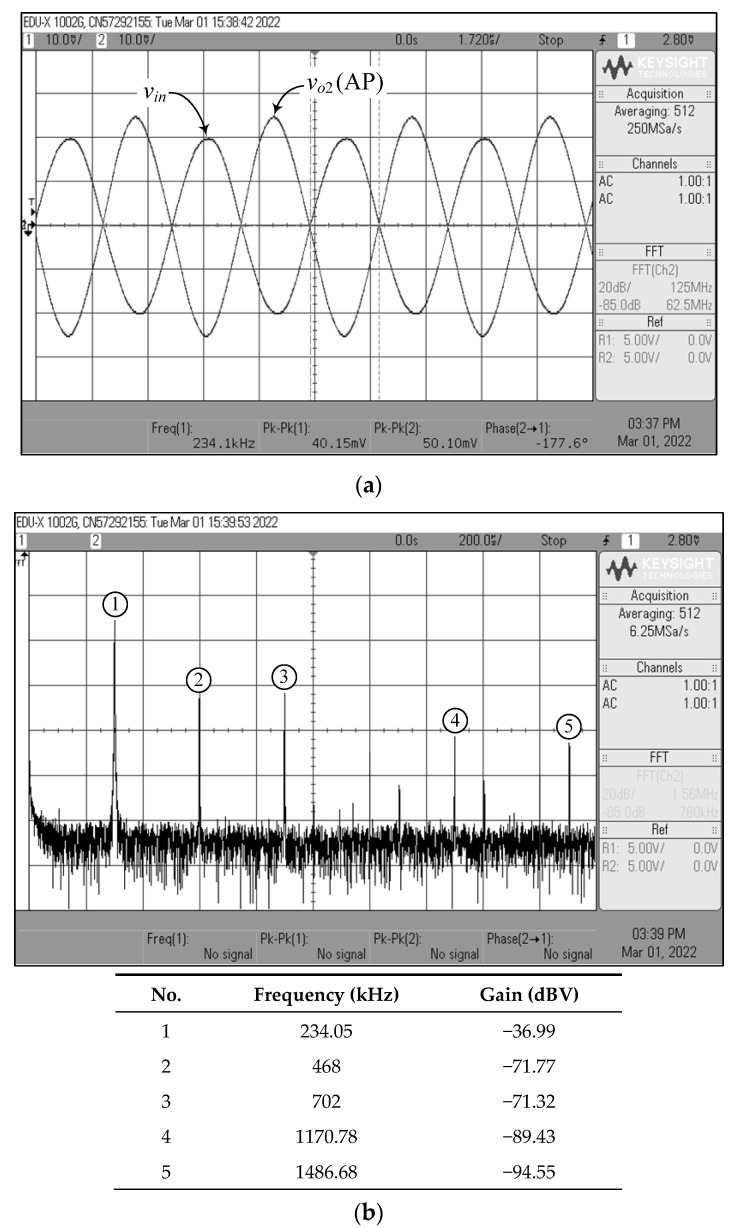
Measured waveforms of the AP filter in VM: (**a**) input and output time responses; (**b**) frequency spectrum.

**Figure 28 sensors-22-05303-f028:**
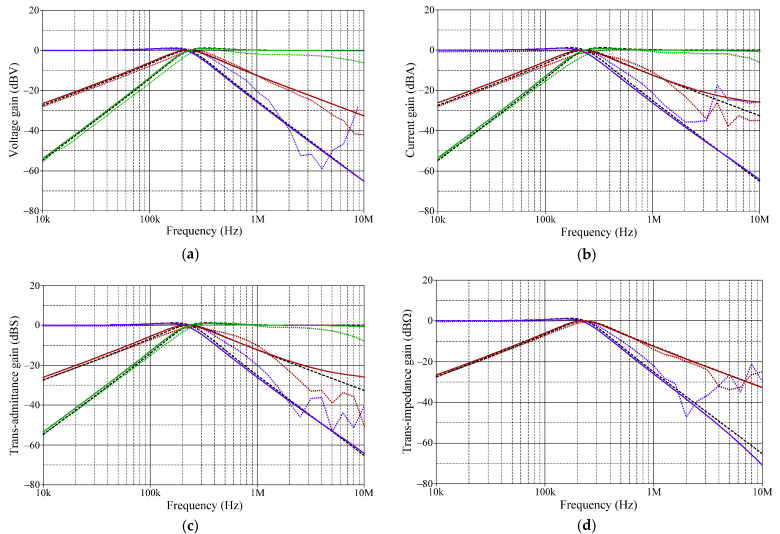
Measured and theoretical frequency responses of the proposed filter (measured in solid line, theoretical in dashed line, simulated in dotted lines): (**a**) LP, BP, and HP responses in VM; (**b**) LP, BP, and HP responses in CM; (**c**) LP, BP, and HP responses in TAM; (**d**) LP and BP responses in TIM.

**Figure 29 sensors-22-05303-f029:**
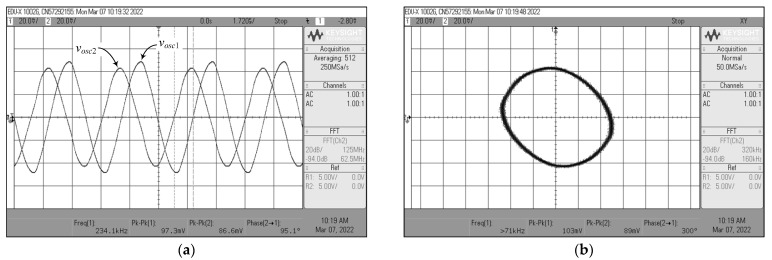
Measured quadrature output voltages *v_osc_*_1_ and *v_osc_*_2_ of Figure 4: (**a**) time-domain waveforms; (**b**) Lissajous pattern.

**Figure 30 sensors-22-05303-f030:**
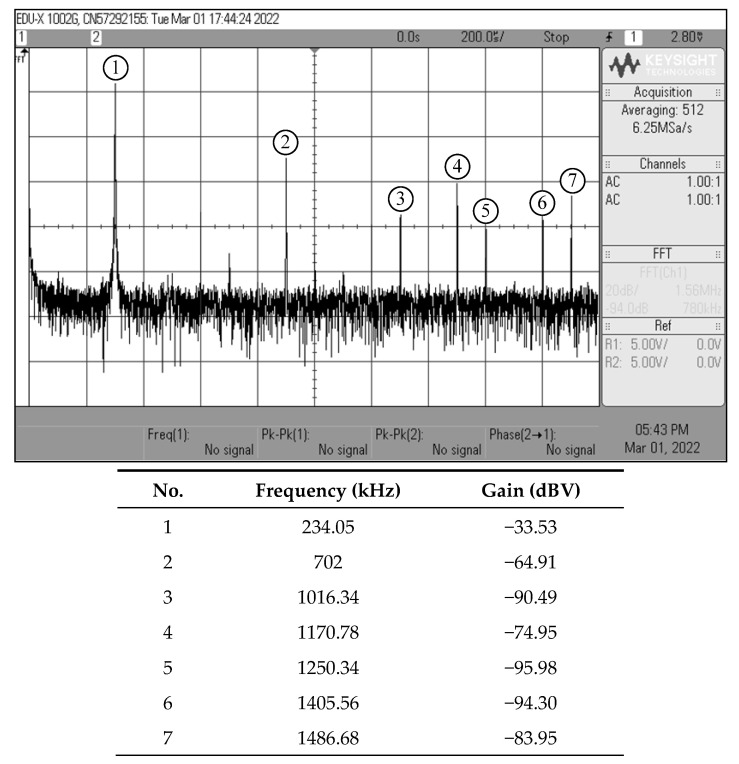
Measured frequency spectrum of *v_osc_*_1_ output.

**Figure 31 sensors-22-05303-f031:**
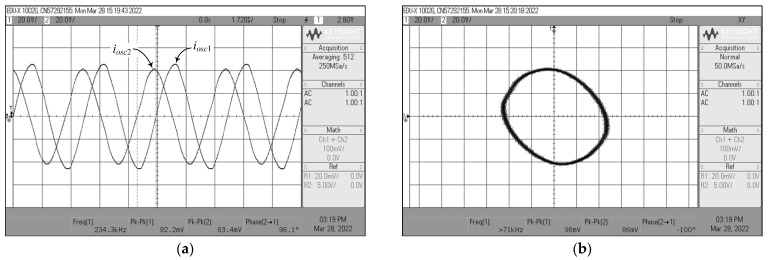
Measured quadrature output currents *i_osc_*_1_ and *i_osc_*_2_ of Figure 4: (**a**) time-domain waveforms; (**b**) Lissajous pattern.

**Figure 32 sensors-22-05303-f032:**
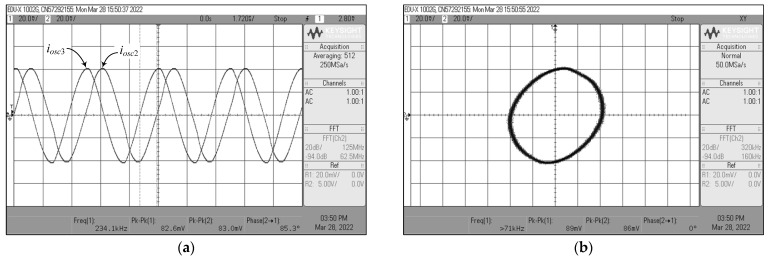
Measured quadrature output currents *i_osc_*_2_ and *i_osc_*_3_ of Figure 4: (**a**) time-domain waveforms; (**b**) Lissajous pattern.

**Figure 33 sensors-22-05303-f033:**
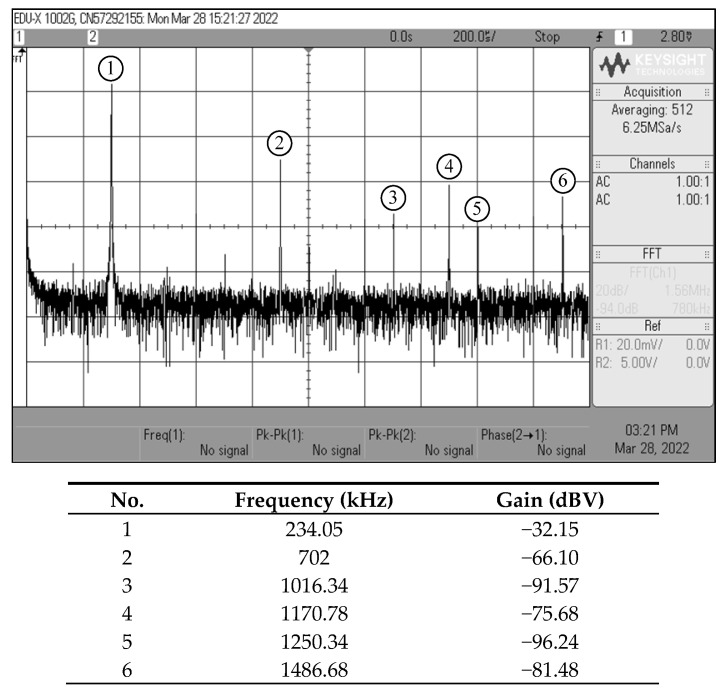
Measured frequency spectrum of *i_osc_*_1_ output.

**Table 1 sensors-22-05303-t001:** Comparison of the proposed circuit to previous related MUBF and QO circuits in [1,2,3,4,5,6,7,8,9,10,11,12,13,14,15,16,17,18,19,20,21,22,23,24,25,26,27,28,29,30,31,32,33,34,35,36,37,38,39,40,41,42,43,44,45,46,47,48,49,50].

Ref./ Year	Working as Both MUBF and QO	No. of Active and Passive Used	MUBF					QO		Inbuilt Tunability	Technology	Supply Voltages (V)	Power Consumption (W)	Technology	Supply Voltages (V)
			Filter Function Realized				Independent Tunable *Q*	Type (VM/CM) and Number of Outputs	Independent Tuning of OC and OF						
			VM	CM	TAM	TIM									
[1]/2003	N	DO-CCCII = 4, C = 3	LP, BP, HP	LP, BP, HP	LP, BP, HP	LP, BP, HP	Y	**--**	**--**	Y	HF3 CMOS	±5	N/A	**--**	**--**
[2]/2006	N	UGC = 8, R = 7, C = 2	all five	all five	**--**	**--**	N	**--**	**--**	N	1.2 μm CMOS	±5, −2.35, −3.54	N/A	**--**	**--**
[3]/2009	N	FDCCII = 1, R = 3, C = 2	all five	all five	BP, HP	all five	Y	**--**	**--**	N	TSMC 0.25 μm	±1.25	N/A	**--**	**--**
[4]/2009	N	DVCC = 3, Rmos = 3, C = 2	LP, BP, BS	all five	all five	LP, BP	Y	**--**	**--**	N	TSMC 0.35 μm	±1.5, 0.75	5.76 m	**--**	**--**
[5]/2009	N	OTA = 5, C = 2	all five	all five	all five	all five	Y	**--**	**--**	Y	TSMC 0.35 μm	±1.65, −1	30.95 m	**--**	**--**
[6]/2009	N	MO-CCII = 3, R = 3, C = 2	**--**	all five	**--**	all five	N	**--**	**--**	N	TSMC 0.18 μm	±1.25, −0.65	N/A	**--**	**--**
[7]/2010	N	OTA = 3, DO-OTA = 1, MO-OTA = 1, C = 2	all five	all five	all five	all five	N	**--**	**--**	Y	TSMC 0.25 μm	±1.25	N/A	**--**	**--**
[8]/2010	N	SCFOA = 1, R = 3, C = 2	all five	LP, BP, BS	**--**	**--**	N	**--**	**--**	N	TSMC 0.25 μm	±1.25	2.53 m	**--**	**--**
[9]/2011	N	DDCC = 3, R = 4, C = 2	all five	all five	all five	all five	Y	**--**	**--**	N	TSMC 0.25 μm	±1.25, +0.41	N/A	**--**	**--**
[10]/2013	N	MO-CCCII = 4, C = 2	all five	all five	all five	all five	N	**--**	**--**	Y	AMS 0.35 μm	±1.25	N/A	**--**	**--**
[11]/2013	N	VDTA = 2, C = 2	all five	**--**	all five	**--**	Y	**--**	**--**	Y	TSMC 0.18 μm	±1.5	N/A	**--**	**--**
[12]/2016	N	FDCCII = 1, DDCC = 1, R = 6, C = 2	all five	all five	all five	all five	Y	**--**	**--**	N	TSMC 0.18 μm	±0.9, ±0.38	N/A	AD844	±15
[13]/2016	N	FDCCII = 2, R = 5, C = 2	all five	all five	all five	all five	N	**--**	**--**	N	TSMC 0.18 μm	±0.9	N/A	**--**	**--**
[14]/2016	N	DP-CCII = 6, MO-CCII = 2, R = 4, C = 2	all five	all five	all five	all five	Y	**--**	**--**	Y	TSMC 0.18 μm	±0.75	3.26 m	**--**	**--**
[15]/2016	N	DPCF = 5, VF = 2, switch = 3, R = 4, C = 2	all five	all five	all five	all five	Y	**--**	**--**	Y	TSMC 0.18 μm	±1.5	1.2 m	**--**	**--**
[16]/2016	N	VDTA = 1, R = 1, C = 3	LP, BP, HP	LP, BP, HP	**--**	**--**	Y	**--**	**--**	Y	TSMC 0.18 μm	±0.9	0.54 m	**--**	**--**
[17]/2017	N	CCCCTA = 3, C = 2	all five	all five	all five	LP, BP, HP	Y	**--**	**--**	Y	TSMC 0.18 μm	±0.9	1.99 m	**--**	**--**
[18]/2017	N	MI-OTA = 3, MO-OTA = 3, C = 2	all five	all five	all five	all five	N	**--**	**--**	Y	TSMC 0.18 μm	±0.5	75 μ	**--**	**--**
[19]/2017	N	DVCC = 1, MO-CCII = 1, R = 4, C = 2	**--**	all five	**--**	all five	Y	**--**	**--**	N	TSMC 0.18 μm	±0.9, ±0.38	N/A	**--**	**--**
[20]/2017	N	OTA = 1, DO-OTA = 3, switch = 1, C = 2	**--**	LP, BP, HP	LP, BP, HP	**--**	Y	**--**	**--**	Y	TSMC 0.35 μm	N/A	1.3 m	**--**	**--**
[21]/2017	N	DXCCDITA = 1, R = 2, C = 2	all five	all five	BP, HP	all five	N	**--**	**--**	Y	TSMC 0.35 μm	±1.5, +0.55	N/A	AD844, LM13700	±5
[22]/2018	N	FDCCII = 2, R = 4, C = 2	all five	all five	all five	all five	Y	**--**	**--**	N	TSMC 0.18 μm	±0.9	1.32 m	**--**	**--**
[23]/2019	N	VCII = 3, I-CB = 1, R = 3, C = 3	all five	all five	all five	all five	N	**--**	**--**	N	TSMC 0.18 μm	±0.9	1.47 μ	**--**	**--**
[24]/2019	N	VD-DXCC = 1, R = 2, C = 2	all five	all five	**--**	**--**	Y	**--**	**--**	Y	PDK 0.18 μm	±1.25	2.237 m	**--**	**--**
[25]/2020	N	OTA = 5, C = 2	all five	all five	all five	all five	Y	**--**	**--**	Y	ADE 0.18 μm	±0.9, −0.36	0.191 m	**--**	**--**
[26]/2020	N	EXCCTA = 2, switch = 1, R = 4, C = 2	all five	all five	all five	all five	Y	**--**	**--**	Y	PDK 0.18 μm	±1.25	N/A	**--**	**--**
[27]/2021	N	VD-EXCCII = 1, R = 3, C = 3	all five	all five	all five	all five	Y	**--**	**--**	Y	PDK 0.18 μm	±1.25	5.76 m	**--**	**--**
[28]/2021	N	EX-CCCII = 1, R = 1, C = 2	all five	all five	all five	BP, HP	N	**--**	**--**	Y	TSMC 0.18 μm	±0.5	1.35 m	AD844	±8
[29]/2021	N	VDBA = 2, R = 2, C = 2	all five	all five	all five	LP, BP	Y	**--**	**--**	Y	TSMC 0.18 μm	±0.75	0.373 m	LT1228	±5
[30]/2022	N	VDBA = 3, R = 1, C = 2	all five	all five	all five	all five	Y	**--**	**--**	Y	PDK 0.18 μm	±1.25	5.482 m	CA3080, LF356	±5
[31]/2022	N	DVCC = 3, R = 4, C = 2	LP, BP, HP	all five	BP, HP	LP, BP, HP	N	**--**	**--**	N	TSMC 0.18 μm	±1.25, +0.55	8.47 m	AD844	±12
[32]/2006	N	FDCCII = 1, R = 3, C = 2	**--**	**--**	**--**	**--**	**−−**	VM/CM, VM = 2, CM = 2	Y	N	TSMC 0.18 μm	±2.5	118.1 m	**--**	**--**
[33]/2009	N	CDTA = 2, R = 1, C = 2	**--**	**--**	**--**	**--**	**−−**	VM/CM, VM = 2, CM = 2	Y	Y	MIETEC 0.5 μm	N/A	N/A	**--**	**--**
[34]/2009	N	DVCC = 3, R = 3, C = 3	**--**	**--**	**--**	**--**	**−−**	VM/CM, VM = 5, CM = 2	Y	N	MIETEC 0.5 μm	N/A	N/A	**--**	**--**
[35]/2014	N	DVCCTA = 1, R = 2, C = 2	**--**	**--**	**--**	**--**	**−−**	VM/CM, VM = 2, CM = 2	Y	Y	TSMC 0.18 μm	±0.9, −0.5	2.283 m	**--**	**--**
[36]/2016	N	CCCTA = 1, C = 2	**--**	**--**	**--**	**--**	**−−**	VM/CM, VM = 2, CM = 2	Y	Y	BJT, TSMC 0.35 μm	±1	N/A	**--**	**--**
[37]/2016	N	VDCC = 2, R = 2, C = 2	**--**	**--**	**--**	**--**	**−−**	VM/CM, VM = 2, CM = 3	Y	Y	TSMC 0.18 μm	±0.9	N/A	**--**	**--**
[38]/2017	N	VDTA = 1, C = 2	**--**	**--**	**--**	**--**	**−−**	VM/CM, VM = 2, CM = 2	Y	Y	TSMC 0.25 μm	±1.5	2.09 m	**--**	**--**
[39]/2020	N	DX- MOCCII = 2, Rmos = 1, R = 2, C = 2	**--**	**--**	**--**	**--**	**−−**	VM/CM, VM = 4, CM = 3	Y	N	TSMC 0.25 μm	±1.25, −0.3, +0.81	6.87 m	AD844	±9.5
[40]/2022	N	VDGA = 1, R = 1, C = 2	**--**	**--**	**--**	**--**	**−−**	VM/CM, VM = 2, CM = 2	N	Y	TSMC 0.35 μm	±1.5	1.36 m	**--**	**--**
[41]/2011	Y	DVCCCTA = 1, C = 2	LP, BP	**--**	**--**	**--**	Y	VM/CM, VM = 2, CM = 2	Y	Y	TSMC 0.25 μm	±1.25	N/A	**--**	**--**
[42]/2014	Y	CDTA = 2, C = 2	**--**	all five	**--**	**--**	N	CM, CM = 2	N	Y	TSMC 0.18 μm	±1.5	N/A	AD844, CA3080	±12
[43]/2014	Y	CDTA = 3, C = 2	**--**	all five	**--**	**--**	N	CM, CM = 4	Y	Y	MIETEC 0.5 μm	±2.5	19.6 m	**--**	**--**
[44]/2017	Y	VDDDA = 3, R = 1, C = 2	all five	**--**	**--**	**--**	Y	VM, VM = 2	Y	Y	TSMC 0.18 μm	±0.9	0.343 m	AD830, LM13700	±5
[45]/2017	Y	VDCC = 2, switch = 3, R = 2, C = 2	**--**	all five	**--**	**--**	Y	VM/CM, VM = 2, CM = 2	Y	Y	TSMC 0.18 μm	±0.9	N/A	OPA860	N/A
[46]/2019	Y	CCFTA = 2, C = 2	**--**	all five	**--**	**--**	Y	VM/CM VM = 2 CM = 4	Y	Y	TSMC 0.18 μm	±1	2 m	**--**	**--**
[47]/2020	Y	CCII = 2, R = 3, C = 2	all five	**--**	**--**	**--**	Y	VM, VM = 2	Y	N	IBM 0.13 μm	±0.75, +0.23	5.03 m	AD844	±6
[48]/2020	Y	VDGA = 1, R = 2, C = 2	LP, BP, HP	LP, BP, HP	**--**	**--**	Y	VM/CM, VM = 2, CM = 2	N	Y	TSMC 0.25 μm	±1	1.49 m	**--**	**--**
[49]/2021	Y	MI-OTA = 3, OTA = 1, C = 2	all five	**--**	**--**	**--**	Y	VM, VM = 3	Y	Y	TSMC 0.18 μm	±1.2	96 μ	LM13700	±5
[50]/2021	Y	VDCC = 2, switch = 2, R = 1, C = 2	**--**	all five	**--**	**--**	N	VM/CM, VM = 2, CM = 2	Y	Y	TSMC 0.18 μm	±0.9	N/A	OPA860	N/A
This work	Y	VDGA = 1, R = 1, C = 2	all five	all five	all five	LP, BP	Y	VM/CM, VM = 2, CM = 3	Y	Y	TSMC 0.18 μm	±0.9	1.31 m	LM13600	±5

Notes: Y = Yes, N = No, N/A = not available, “**--**” = not realized, R = resistor, C = capacitor, Rmos = MOS-based electronic resistor, OTA = operational transconductance amplifier, DO-OTA = dual-output OTA, MO-OTA = multiple-output OTA, MI-OTA = multiple-input OTA, CCII = second-generation current conveyor, MO-CCII = multiple-output CCII, DP-CCII = digitally programmable current conveyor, DO-CCCII = dual-output second-generation current-controlled conveyor, MO-CCCII = multiple-outputs current-controlled conveyor, FDCCII = fully differential current conveyor, CFOA = current feedback operational amplifier, SCFOA = specific CFOA, UGC = unity-gain cell, DVCC = differential voltage current conveyor, DDCC = differential difference current conveyor, DVCCTA = differential voltage current conveyor transconductance amplifier, DVCCCTA = differential voltage current-controlled conveyor transconductance amplifier, CCCCTA = current controlled current conveyor transconductance amplifier, VDTA = voltage differencing transconductance amplifier, VDGA = voltage differencing gain amplifier, DPCF = digitally programmable current follower, VF = voltage follower, DXCCDITA = dual X current conveyor differential input transconductance amplifier, VCII = second-generation voltage conveyor, I-CB = inverting current buffer, VD-DXCC = voltage differencing dual X current conveyor, EXCCTA = extra X current conveyor transconductance amplifier, VD-EXCCII = voltage differencing extra X CCII, EX-CCCII = extra X CCCII, VDBA = voltage differencing buffered amplifier, CDTA = current differencing transconductance amplifier, CCFTA = current-controlled current follower transconductance amplifier, VDDDA = voltage differencing differential difference amplifier.

**Table 2 sensors-22-05303-t002:** Aspect ratios of the CMOS transistors of the VDGA in Figure 2.

Transistors	*W* (μm)	*L* (μm)
M_1*k*_–M_2*k*_	23.5	0.18
M_3*k*_–M_4*k*_	30	0.18
M_5*k*_–M_7*k*_	5	0.18
M_8*k*_–M_9*k*_	5.5	0.18

**Table 3 sensors-22-05303-t003:** Simulated *f_o_* and corresponding percentage errors of the proposed mixed-mode universal biquad filter in Figure 3, where theoretical *f_o_* = 3.18 MHz.

		LP	BP	HP	BS	AP
VM	*f_o_* (MHz)	3.098	3.105	3.064	2.999	3.030
	Error (%)	2.579	2.371	3.638	5.686	4.714
CM	*f_o_* (MHz)	3.099	3.106	3.068	2.964	3.010
	Error (%)	2.547	2.336	3.522	6.786	5.346
TAM	*f_o_* (MHz)	3.100	3.104	3.067	2.964	3.009
	Error (%)	2.525	2.406	3.557	6.786	5.377
TIM	*f_o_* (MHz)	3.100	3.106	**−**	**−**	**−**
	Error (%)	2.519	2.343	**−**	**−**	**−**

**Table 4 sensors-22-05303-t004:** THDs and DC components of VM, CM, TAM, and TIM outputs with 3.18 MHz sinusoidal input signal.

		LP	BP	HP	BS	AP
VM	THD (%)	0.47	0.45	0.55	1.92	1.29
	DC component (mV)	9.55	2.36	2.37	2.46	3.04
CM	THD (%)	1.5	1.49	0.9	1.87	1.39
	DC component (μA)	10.79	4.42	0.044	10.74	15.17
TAM	THD (%)	1.57	1.45	0.9	1.86	1.26
	DC component (μA)	10.74	4.43	0.015	10.72	15.15
TIM	THD (%)	0.58	0.38	**−**	**−**	**−**
	DC component (mV)	9.62	2.39	**−**	**−**	**−**

## Data Availability

The data supporting the results presented in this work are available on request from the authors.

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
