# Peer review of "Single VDGA-Based Mixed-Mode Universal Filter and Dual-Mode Quadrature Oscillator"

_sensors, 2022, doi:10.3390/s22145303_

Round 1

Reviewer 1 Report

This paper proposes a circuit for a mixed-mode universal biquadratic filter and a dual-mode quadrature oscillator. They require a single voltage differencing gain amplifier, one resistor and two capacitors. The mixed-mode universal filter has three inputs and two outputs. It can perform standard biquadratic filter functions in voltage-, current-, trans-admittance-, and trans-impedance-mode. The dual-mode quadrature oscillator allows for orthogonal resistive and/or electronic control of the oscillation condition and the oscillation frequency. Simulations and experimental results are presented to support the findings of the paper.

The paper is well-written and interesting for the purpose of the Sensors journal.

I suggest a revision based on the following points in order to improve the quality of the paper:

1.- Introduction section. Line 36, it seems excessive to refer 40 references in a row [1–40], please develop. Similar issues are found in lines 45, 47, 48.

2.- All equations in the manuscript must be referred or demonstrated.

3.- I suggest to add a nomenclature section.

4.- Results presented in figures 6 to 15 must be discussed.

5.- Experimental results shown in the figures of Section 7 must be superimposed with simulation results.

6.- Results presented in Figs. 28 require a detailed discussion, clearly indicating the origin of the discrepancies.

I hope this revision can help the authors to improve the quality and readability of the paper.

Reviewer 2 Report

I have few questions prior to accept it. 

1. Abstract must contain key findings of the work. 

2. Provide most  relevant keywords.

3. Introduction must contain clear objective of the work along with motivation. 

4. Provide some latest references and compare your result with better number of datasets. 

Reviewer 3 Report

This paper presents an active mixed-mode universal biquadratic filter (MUBF) and dual-mode quadrature oscillator (DMQO) circuit designed using a single voltage differencing gain amplifier (VDGA), extensive simulations are presented to prove the study hypothesis. A series of experimental tests have been carried out with the integrated circuits LM13600. Author claims: “To further demonstrate the practicability of the circuit, the experimental test results using commercially available ICs are also conducted.” Therefore, the simulation results presented in section 6, evaluating different ratios for the TSMC 0.18-um CMOS technology transistors, were not realized or fabricated.

 In order to illustrate the different mixed-modes of the universal filter and the dual mode quadrature oscillator, the author provides several numerical plots. PSPICE simulations are presented for validating the new structure for a MUBF and a DMQO. According to the author, the novelty of the proposal lies in using a single VDGA to implement a MUBF and DMQO.

 I do, however, believe that we need to address three important issues:

 1. Why in Figures 23-33 did you put 234.05 MHz? The frequency measured on the oscilloscope is in kHz.

2. The results presented in section “Results” are simulations using PSpice. I would like to see the demonstration of the results in an electronic implementation by using the 0.18-um CMOS technology.

3. Add a table comparing state-of-the-art MUBF and a DMQO simulated using CMOS technology but not fabricated with it in the "Discussion" section. Present only papers published in Clarivate-indexed journals.

In my opinion, the manuscript is incomplete and needs to be reworked.

Round 2

Reviewer 1 Report

The authors have replied my concerns

Reviewer 3 Report

I recommend accepting the manuscript. All of my suggestions were taken into account by the authors.